# Functional proteoform group deconvolution reveals a broader spectrum of ibrutinib off-targets

Isabelle Rose Leo [1], Elena Kunold[1,2], Anastasia Audrey [3], Marianna Tampere[4], Jürgen Eirich [5], Janne Lehtiö [1] & Rozbeh Jafari [1] ✉

Proteome-wide profiling has revealed that targeted drugs can have complex protein interaction landscapes. However, it's a challenge to profile drug targets while systematically accounting for the dynamic protein variations that produce populations of multiple proteoforms. We address this problem by combining thermal proteome profiling (TPP) with functional proteoform group detection to refine the target landscape of ibrutinib. In addition to known targets, we implicate additional specific functional proteoform groups linking ibrutinib to mechanisms in immunomodulation and cellular processes like Golgi trafficking, endosomal trafficking, and glycosylation. Further, we identify variability in functional proteoform group profiles in a CLL cohort, linked to treatment status and ex vivo response and resistance. This offers deeper insights into the impacts of functional proteoform groups in a clinical treatment setting and suggests complex biological effects linked to off-target engagement. These results provide a framework for interpreting clinically observed off-target processes and adverse events, highlighting the importance of functional proteoform group-level deconvolution in understanding drug interactions and their functional impacts with potential applications in precision medicine.

Proteins are responsible for execution of cellular processes and act as critical mediators of phenotypic traits. Although the human genome encodes approximately 20,000 unique protein sequences, a multitude of processes including alternative splicing, post-translational modification (PTM), and proteolytic cleavage serve to significantly expand this diversity, generating proteoforms in numbers several orders of magnitude higher[1]. Additionally, important differences such as biomolecular interaction state, solute accessibility, or subcellular localization may lead to physically and functionally distinct but not necessarily chemically distinct proteoform entities. Collectively, these functional proteoform groups open the door to a much wider range of function and evolution of adaptive phenotypes under selective

pressure[1]. Additionally, many diseases, such as cancer, have been linked to changes that result in alterations of proteoforms, such as decay of aberrant transcripts[2], splicing dynamics[3], translation[4], and post-translational modifications[5]. With highly variable proteomes that are under continuous selective pressure, cancers demonstrate numerous examples where protein complex composition[6–8] and proteoform drift[9–12] play important roles in biology and therapeutic response. Therefore, targeting cancer-associated proteoforms has emerged as an area of intense clinical interest, notably in pediatric cancers[13], prostate cancer[14], melanoma[15,16], and breast cancer[17]. This highlights the importance of identifying and characterizing different proteoforms, as well as developing drugs that target them. These

[1]Clinical Proteomics Mass Spectrometry, Department of Oncology-Pathology, Karolinska Institutet, Science for Life Laboratory, Solna, Sweden. [2]Evotec International GmbH, München, Germany. [3]Department of Medical Oncology, University Medical Center Groningen, Groningen, the Netherlands. [4]Precision Cancer Medicine, Department of Oncology-Pathology, Karolinska Institutet, Science for Life Laboratory, Solna, Sweden. [5]Institute of Plant Biology and Biotechnology, University of Münster, Münster, Germany. ✉e-mail: rozbeh.jafari@ki.se

efforts have the potential to significantly contribute to advancing precision medicine and delivering on the promise of personalized therapy.

In this context, development of targeted therapies for treatment of cancer has been one of the major breakthroughs in the field. These advances have been fueled by the improved ability to connect a disease phenotype to a specific causative protein, and then translate these insights into a drug that targets that protein to correct its function and restore the phenotype. Although the focus of targeted drug discovery is on development of drugs that affect only a single target, global mapping of drug-proteome interactions have revealed that many approved targeted therapies affect more than a single target protein[18,19]. These off-target effects can often help rationalize clinical observations both with respect to efficacy and toxicity. However, the full understanding of how drugs bind and affect different proteoforms has been missing, given the difficulty of conducting proteoform-level analysis on a high-throughput scale.

In general, proteoform species are challenging to detect and quantify because they lack a one-to-one relationship with genetic information, making their existence hard to predict and anticipate. Additionally, proteoforms may occur at low quantitative levels, or only transiently. Thus, molecular methods for proteoform characterization have to allow for unbiased identification without a need to pre-define or isolate variants, and offer high sensitivity and dynamic range. Targeted proteomics techniques (including parallel reaction monitoring, immuno- and proximity ligation assays) can support platforms for flexible protein characterization with high sensitivity[20]. But to effectively characterize proteoforms, these methods must be set up with a robustly pre-defined protein target or epitope. In the field of mass spectrometry (MS)-based proteomics, both top-down and bottom-up approaches have been used to conduct proteoform analysis without a need to pre-define variants. For example, top-down proteomics has been employed to assemble human proteoform reference maps[21,22]; however, this method currently has limitations in depth and throughput[22]. To address this issue, bottom-up proteomics uses digested peptide libraries to achieve the broadest range of identifications. This makes it suitable for global proteome detection approaches[23], but at the cost of inferring proteoforms rather than identifying them directly. The term "functional proteoform group" specifies these inferred proteoforms, where data supports a proteoform-level distinction but does not necessarily demonstrate a unique, specific proteoform[24].

Many methods for global proteoform inference have been developed, including using abundance profiles[24–26] and thermal proteome profiling (TPP)[27]. TPP is a powerful method for systematic detection and annotation of functional aspects of the proteome that uses a series of temperature treatments to resolve proteins based on their thermal stability[28] which we have previously extended for functional proteoform group identification[27]. The unique thermal stability of proteoforms can e.g., arise from different sets of PTMs, alternative splicing or proteolytic processing, and interactions with proteins, DNA, RNA, metabolites or drugs, therefore TPP approaches can capture many proteoform types[27,29]. Furthermore, TPP can be applied to a range of biological systems and used to analyze functional proteoform groups in their natural contexts[27].

Here, we use TPP to describe small molecule drug interactions with functional proteoform groups. We map the target landscape of ibrutinib, a clinically used Bruton tyrosine kinase (BTK) inhibitor[30–32]. Our choice to focus on ibrutinib is motivated by clinical data that point to a complex relationship between on-target binding, efficacy, and toxicity. For example, ~85% of chronic lymphocytic leukemia (CLL) patients treated with ibrutinib develop BTK pathway mutations[33–35]. However, some of these patients continue to respond to ibrutinib[36,37], which could be attributed to a range of factors, including possibly inhibition of additional targets. Although previous proteomics studies have identified a wide range of ibrutinib off-targets[18,38], which could be clinically beneficial and useful for repurposing[39,40] or harmful to patients[41,42], many clinical observations remain to be rationalized.

Therefore, we hypothesize that analysis of functional proteoform groups can provide a more complete picture of the ibrutinib target landscape and extend our understanding of treatment sensitivity, off-target events, and resistance mechanisms. Our study reveals additional targets for ibrutinib, including functional proteoform groups involved in Golgi trafficking, glycosylation, cell adhesion, and endosomal processing, as well as some that may amplify drug efficacy and enable BTK-independent immunomodulation.

## Results

### Functional proteoform group identification in ibrutinib-treated cell lysates

To reduce the risk of ambiguity in distinguishing drug targets from secondary protein stability changes, we performed TPP in cell lysates. We used lysates from two different cell lines, a precursor-B acute lymphoblastic leukemia cell line with a somatic form of *BTK*, RCH-ACV[43] (RRID: CVCL_1851), and the adrenocortical carcinoma cell line SW13 also with somatic *BTK*[43] (RRID: CVCL_0542). Lysates in the presence of ibrutinib or equivalent volume of dimethyl sulfoxide (DMSO) were prepared in technical replicates for a total of eight melt curve sets. Each set was thermally denatured in a ten-point tandem-mass tag (TMT) multiplexed temperature curve and pre-fractionated using high-resolution isoelectric focusing[44] for in-depth peptide detection by MS proteomics. In total, 175,379 unique peptides mapping to only one gene symbol were detected from 11,043 gene symbol stratified proteins. After functional proteoform group clustering analysis[27] (Fig. 1a) of these peptides across the whole dataset, 16,079 functional proteoform groups were identified, representing peptide groups with similar behavior in TPP. These thermally inferred proteoforms were investigated for differential melting using nonparametric analysis of response curves (NPARC)[45], considering inferred proteoforms detected in all samples first across the entire dataset and then within each lineage separately (Fig. 1b, Supplementary Fig. 1A). Together, these analyses identified 2305 thermally impacted proteoforms from 1936 gene symbols at a $p$-value threshold of 0.05 (Supplementary Data 1), and 251 from 230 gene symbols at a Benjamini-Hochberg adjusted $p$-value threshold (pAdj) of 0.05.

To validate our approach, we first examined whether TPP coupled with MS proteomics was able to identify BTK, the primary target of ibrutinib. BTK was only identified in the RCH-ACV lysates, which is consistent with the lineage specificity of BTK. We identified two BTK functional proteoform groups which had different baseline melting behavior (BTK_1 and BTK_2) (Fig. 1c), and both were stabilized in the presence of ibrutinib (Supplementary Fig. 1B). Although BTK_1 was the only BTK functional proteoform group that met the pAdj < 0.05 NPARC test significance threshold, BTK_2 was shifted based on a $p < 0.05$ threshold (pAdj = 0.22). BTK_1 contained several peptides derived from the ATP binding site (Fig. 1d, Supplementary Fig. 1D) that were not detected in BTK_2, which suggests that interactions, modifications, conformational changes, or splice variants reduced relative representation of these peptides in BTK_2. We did not observe cysteine 481 (C481), the specific residue that is covalently targeted by ibrutinib, in either functional proteoform group. We used BTK results to calibrate significance levels for further result interpretation, and we proceeded to consider pAdj < 0.05 as likely thermally impacted. Although results such as BTK_2 meeting only a $p < 0.05$ threshold could be false positives, they can not be excluded. For transparency, we noted these cases may be plausibly thermally impacted and included them in Supplementary Data 1, to ensure false negatives were not implied. In addition to BTK, previous studies[18] have revealed that ibrutinib binds a wide range of proteins (Supplementary Data 2), and our analysis confirmed several of these (Supplementary Fig. 1C).

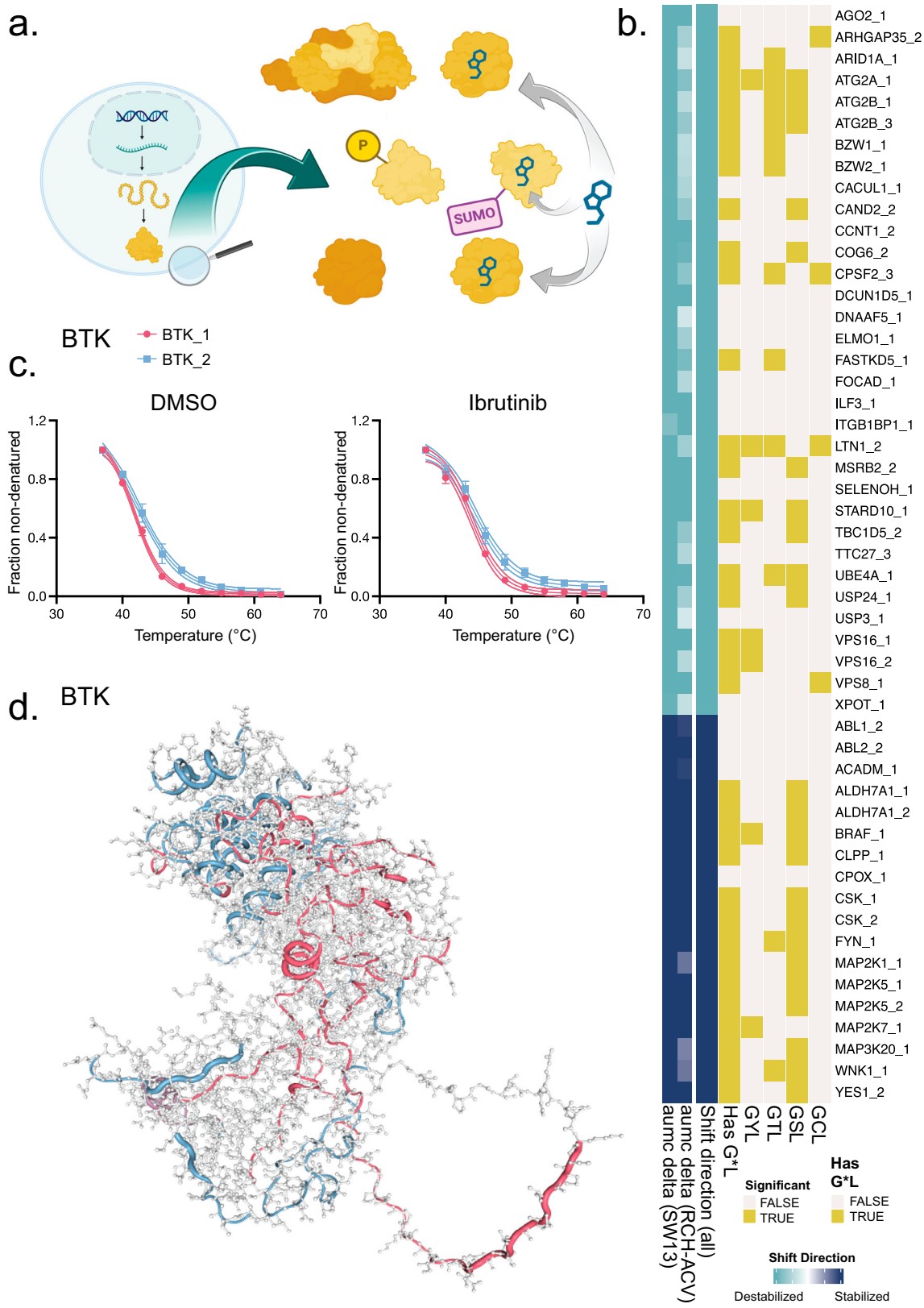

Ibrutinib's C481 BTK binding site is flanked by G and L residues, and this sequence has been profiled in other off-target studies and is known to be homologous in off-targets[38]. Although GCL enrichment was not significant, we confirmed that ibrutinib targets in the $p < 0.05$ population display G*L motif enrichment in a 2-sample equality of proportion test with Yates continuity correction, with notable over-representation of amino acids that can be modified by phosphorylation flanked by G and L residues (GCL: $p = 0.169$, GSL: $p = 0.00089$, GTL: $p = 0.00014$, GYL: $p = 0.224$, any G*L, $p = 0.00034$) (top hits annotated in Fig. 1b). Taken together, these results indicate that TPP coupled with MS-based proteomics was sensitive to thermal differences between drug and DMSO conditions. Overall, the fact that we identified BTK as well as other known off-targets of ibrutinib serves as internal validation of our experimental setup.

**Fig. 1 | Characteristics of top ibrutinib binding candidates and BTK proteoforms. a** Illustration of the functional proteoform group concept, seeking to pair drug interaction information with specific study of protein variation lacking a direct parallel with genetic information. Functional proteoform groups can include multiple chemical or physical proteoform types that have potential for differential thermal stability and differential drug interactions. Created with BioRender.com. **b** The top 51 functional proteoform group results differentially melting in ibrutinib-treated versus untreated lysates, as determined by NPARC analysis, with highlighted GCL, GSL, GTL, or GYL sequences. Selected proteoforms meet a Benjamini-Hochberg adjusted *p*-value threshold of less than 0.0001 for models with detection in all cell lines. **c** Fraction non-denatured for each BTK functional proteoform group detected in RCH-ACV, colored by proteoform group membership assignment demonstrating melting behavior by functional proteoform group and treatment status, which illustrates the relative thermal stability difference which is seen in both ibrutinib and vehicle melt curves. The melt curves for each condition represent *n* = 10 temperatures each, each of the 10 temperature measurements performed in technical duplicates for each treatment condition and detected only in RCH-ACV. Melt curves are presented along with the 4PL curve fit and 95% confidence interval of the fitted model, and each point shows the mean fraction non-denatured and ±standard deviation (SD). **d** Structural diagram of peptide mappings, generated using the Alphafold structure for BTK generated using the canonical FASTA sequence (sp|Q06187|BTK_HUMAN), showing tube overlays for peptides colored by their functional proteoform group assignments. Regions with multiple matched assignments are displayed with blended translucent coloring, and regions without assigned peptides appear as a gray amino acid backbone. Source data are provided as a Source Data file.

## Using proteoform group data to detect the effects of drug binding on protein-protein interactions and complex formation

In addition to inhibiting activity of a target, drug binding may also lead to conformational changes or allosteric effects that change how the target interacts with its binding partners[46]. To identify cases where thermal profiles indicated disruption of protein complexes, we performed an over-representation analysis (ORA) to identify proteins that form complexes in the CORUM database[47]. Using false discovery rate (FDR)-adjusted hits (pAdj < 0.05), we identified twelve complexes (Fig. 2a, Supplementary Data 3). Among these complexes, three hits (CORVET, HOPS, class C VPS complex) (Fig. 2a, b, Supplementary Figs. 2, 3) share common components and have known interconnected biological functions as membrane tethering complexes[48] and as coordinators of signal transduction[49]. Of particular relevance, this signaling includes NF-kB and AP1, which are induced during immunostimulation[49], and may represent a possible secondary route towards maintaining ibrutinib drug effects. Another identified complex, NUMAC (Supplementary Fig. 4), integrates chromatin remodeling and histone methylation and plays a role in fine-tuning gene expression during heart, lung, and immune cell development[50,51]. The critical histone modifying component of NUMAC, CARM1, has been proposed as a cancer target and its knockdown enhances antigen-induced proliferation and cytotoxicity in tumor-infiltrating T-cells[50], which is also observed in ibrutinib-treated patients[52,53]. Among other affected complexes, the p21(ras)GAP-FYN-LYN-YES complex, the CD20-LCK-LYN-FYN-p75/80 complex and the BRAF-MAP2K1-MAP2K2-YWHAE complex (Supplementary Figs. 5–7), also have functional links to immunostimulatory signaling.

Protein-protein interactions are highly dependent on cell lineage[54], and we observed that the baseline thermal stabilities of complex-associated functional proteoform groups and magnitude of drug-induced thermal changes were not uniform between cell lines. For example, the membrane tethering complexes were only indicated in SW13 (Supplementary Figs. 2, 3), and two of the immunostimulatory signaling complexes only in RCH-ACV (Supplementary Figs. 5,6). To examine this further, we repeated the ORA tests within each cell line separately (Supplementary Data 3) to allow the ORA to have proper statistical input for background detection and because cell line backgrounds affect baseline melting more than drug effects. This indicated several SW13-specific results including the multisynthetase complex, the EARP tethering complex, and the EIF2B2-EIF2B3-EIF2B4-EIF2B5 complex. These observations underscore that cell line-specific predominance of certain complexes could influence the range of interactions and scope of functional drug influence. Despite the cell type specificity of protein complex results, at functional proteoform group-level, thermal changes still occurred for individual components across both cell lines, such as the HOPS complex (Fig. 2b, Supplementary Data 1). This implicates individual functional proteoform group targets from enriched complexes, even where the protein complexes themselves are not indicated.

Our analysis also identified complexes without clear functional links to the known on-target effects of ibrutinib. For example, the COG complex is essential in intra-Golgi transport and glycosylation of proteins and lipids[55], and components were thermally impacted by treatment in both cell lines (Fig. 2a, Supplementary Fig. 8). This complex is primarily found as an octamer in the cytosol, and components are also integrated into numerous other subcomplexes[56], which were also detected in the protein complex ORA (Supplementary Data 3). Although the link between ibrutinib and COG components is not known, changes in immunoglobulin glycosylation and secretion have been identified in ibrutinib-treated CLL patients[57], in agreement with our results.

Another hit without a clear functional link to on-target pathways was the WASH complex (Supplementary Fig. 9). This complex facilitates endosomal trafficking, surface receptor recycling[58] and maintenance of phagocytosis[59]. Ibrutinib treatment is reported to cause defects in both receptor recycling[60], and endosomal trafficking[61,62], particularly of importance in mechanisms of aspergillosis susceptibility and immune cell egress. In addition to these potential links, the WASH complex modulates platelet function through reducing αIIbβ3 integrin cell surface expression[63], which mirrors a BTK-independent ibrutinib effect on αIIbβ3 integrin cell surface levels[64].

To further map the pathways affected by ibrutinib, as indicated by perturbations in functional proteoform groups, we used the BioGRID protein-protein interaction database[65] and performed enrichment analysis, using network topology analysis with network retrieval prioritization[66] (Supplementary Fig. 10, Supplementary Data 4). Input hits were gene symbol IDs with at least one functional proteoform group below the pAdj < 0.05 threshold for at least one cell type. As expected, we identified the B cell receptor signaling target pathway, in alignment with the intended function of ibrutinib; however, we also identified that ibrutinib had BTK-independent effects, including protein autophosphorylation, RNA localization, and cell adhesion. Consistent with the ORA results, other top hits included organelle membrane fusion and Golgi organization. Collectively, these analyses showcase how functional proteoform group analysis of drug effects can reveal specific changes at the level of protein-protein interactions and complex formation. In the specific case studied here, observed effects suggest potential, BTK-independent link between ibrutinib, and COG and WASH complexes, and their respective cellular functions.

## Functional proteoform group analysis enables more nuanced target identification

Although we observed good agreement between our studies and previously reported ibrutinib off-target identification using kinobeads[18] (Supplementary Data 2), we also observed some additional hits, including several examples of clinically highly relevant targets. For example, although not seen as a target in the kinobead study[18], we observed that BRAF had two functional proteoform groups in our

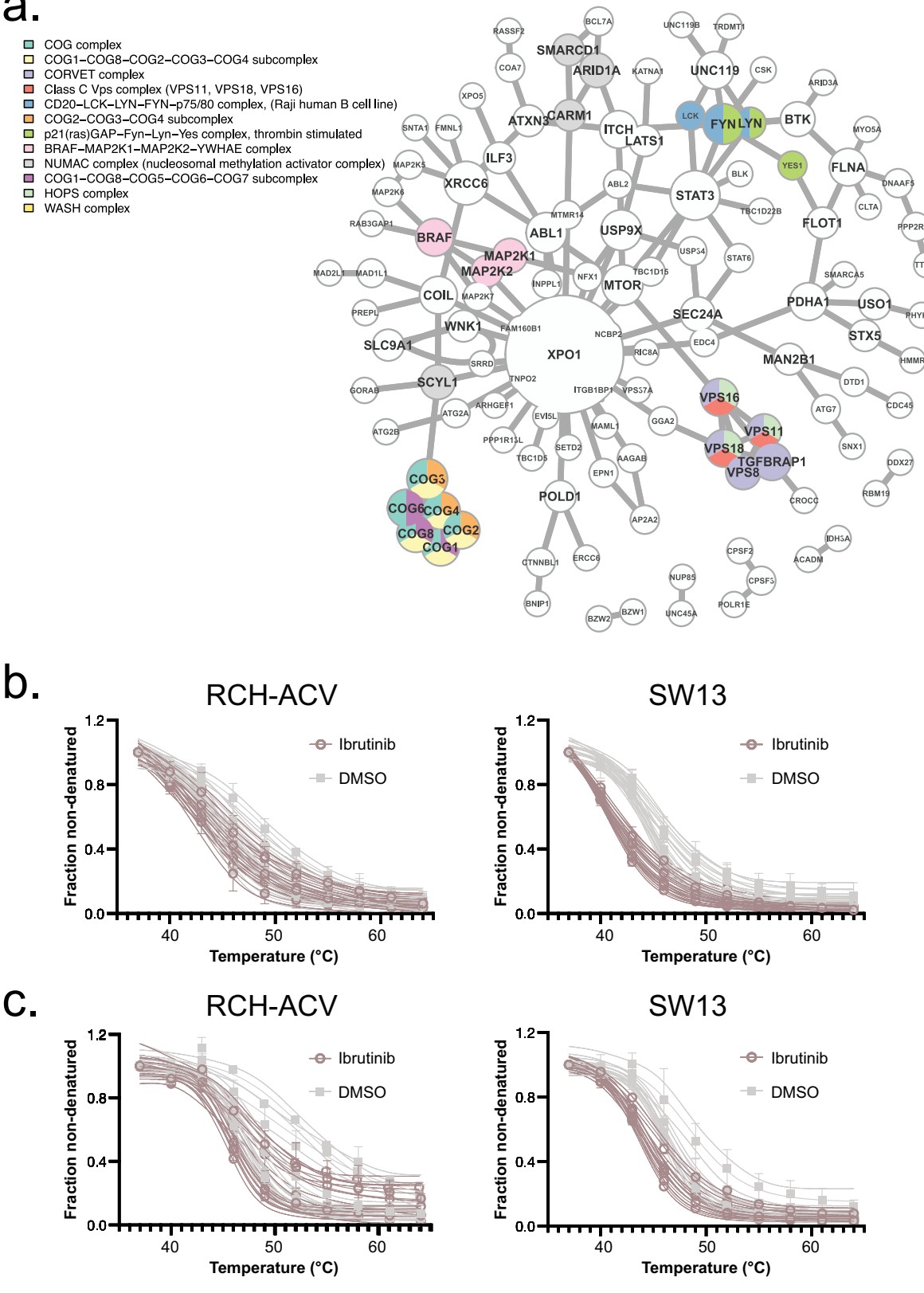

**a.**

Legend:
- COG complex
- COG1–COG8–COG2–COG3–COG4 subcomplex
- CORVET complex
- Class C Vps complex (VPS11, VPS18, VPS16)
- CD20–LCK–LYN–FYN–p75/80 complex, (Raji human B cell line)
- COG2–COG3–COG4 subcomplex
- p21(ras)GAP–Fyn–Lyn–Yes complex, thrombin stimulated
- BRAF–MAP2K1–MAP2K2–YWHAE complex
- NUMAC complex (nucleosomal methylation activator complex)
- COG1–COG8–COG5–COG6–COG7 subcomplex
- HOPS complex
- WASH complex

**b.** RCH-ACV / SW13

**c.** RCH-ACV / SW13

dataset (Fig. 3a, b) and was also a component of a complex flagged in our ORA results (Supplementary Fig. 7). One functional proteoform group was a significant hit in both cell lines individually and across the entire dataset (Supplementary Data 1); however, the other, more thermally stable proteoform did not exhibit significant changes upon ibrutinib treatment for either cell line (Fig. 3b). When BRAF results are taken in aggregate on a gene symbol level, the results indicate that BRAF is stabilized overall (Fig. 3c), which illustrates that aggregation of this kind could mask the more nuanced behavior captured by the functional proteoform group analysis (Fig. 3a, b). Therefore, this suggests that BRAF may exist in distinct forms, with critical differences that may have functional implications.

**Fig. 2 | Physically associated and functionally implicated ibrutinib binding candidates. a** Network plot showing the CORUM complex composition of the sub-network of top NPARC hits and their associations according to the BioGRID interaction database. Nodes are plotted by size according to connectivity, and colored labels indicate membership in an enriched CORUM complex. **b** Functional proteoform group melting behavior for the HOPS complex, showing results flagged in the enrichment analysis input where at least one individual cell line NPARC result showed a shift meeting a significance threshold of $p < 0.05$, which include: VPS11_1, VPS11_2, VPS16_1, VPS16_2, VPS18_1, VPS18_2, VPS33A_1, and VPS33A_2. The melt curves for each condition represent $n = 10$ measurements each, each of the 10 temperature measurements performed in duplicates for each experimental condition and cell line, and are plotted with 95% confidence intervals of 4PL curve fits. Each point shows the mean fraction non-denatured and error bars show ±SD. The complex enrichment result was significant when evaluated using data from both cell lines, pAdj = 0.0392. **c** Functional proteoform group melting behavior for the WASH complex, showing flagged in the enrichment analysis input where at least one individual cell line NPARC result showed a shift meeting a significance threshold of $p < 0.05$, which include: WASHC2C_1, WASHC4_1, WASHC4_2, WASHC4_3, WASHC5_1, WASHC5_2, and WASHC5_3. The melt curves for each condition represent $n = 10$ measurements each, each of the 10 temperature measurements performed in duplicates for each experimental condition and cell line, and are plotted with 95% confidence intervals of 4PL curve fits. Each point shows the mean fraction non-denatured and error bars show ±SD. The complex enrichment result was significant when evaluated using data from both cell lines, pAdj = 0.0392, and in SW13, pAdj = 0.0348. Source data are provided as a Source Data file.

Here, several lines of evidence support this possibility. On one hand, ibrutinib has been investigated for resensitizing BRAF-inhibitor refractory melanomas[67], a benefit not replicated with other BTK inhibitors indicative of ibrutinib-specific off target effects. Additionally, CORUM over-representation results indicated that ibrutinib treatment may interrupt BRAF interaction with MEK proteins (Fig. 2a, Supplementary Fig. 7), and experimental structure of the BRAF kinase domain in complex with MEK[68] appeared to be consistent with the mapping of BRAF_1 peptides (Fig. 3d). On the other hand, the BRAF_2 functional proteoform group may represent a dimerized form of BRAF. It is well established that BRAF is activated by RAS-dependent dimerization[69], including with other RAF kinases (ARAF and CRAF (RAF1)). Our results showed that RAF1 and ARAF were not thermally impacted in any sample or proteoform group, leading us to propose that the proteoform group of BRAF that appears insensitive to ibrutinib are the hetero- or homodimers (Supplementary Fig. 11A, B). Furthermore, dimerized BRAF may be the main population of BRAF in cells, potentially explaining why kinobeads did not identify BRAF as ibrutinib target[18].

The identification of results with mixed drug binding affinity between proteoform group sub pools suggests that these gene symbol IDs would be harder to replicate in a traditional analysis. To probe this generalization further, we examined replication across the full dataset. Among the 217 pAdj < 0.05 gene IDs that were not replicated in the kinobeads study, 8.3% were hits for all proteoform groups, with the rest having a proteoform group above $p = 0.05$ or not clustered into proteoform groups. But among the twelve kinobeads replicated gene IDs, 42% were clustered into proteoform groups and were significant for all proteoform groups. This demonstrates that results previously identified at gene symbol level[18] were proportionately ~5 times more likely to be thermally impacted for all clustered proteoform groups, compared to unreplicated gene symbols.

Thermal proteome profiling may generally be a more sensitive approach for certain drug contexts. However, not all replicated hits from the kinobeads study would have been clear without proteoform group clustering. For example, we identified two functional proteoform groups of the tyrosine kinase YES1, an important member of the src kinase family. YES1_2 was stabilized in RCH-ACV and across the whole dataset, but YES1_1 was not (Fig. 3e). Without functional proteoform group clustering, in the gene symbol aggregated full dataset, YES1 fell outside the significance threshold at $p = 0.0617$, pAdj = 0.433 (Fig. 3f). Structurally, the stabilized YES1_2 peptides mapped in the N-terminal region, which in src kinases is known as the src N-terminal regulatory element (SNRE), an understudied and intrinsically disordered region thought to perform lipid binding, enacting regulatory functions and enabling cell type-specific roles[70] (Supplementary Fig. 11d). These context-dependent lipid interactions could be another mechanism introducing melting variance between samples. The apparent baseline melting difference between RCH-ACV and SW13 is notable, despite a lack of mutations in this protein detected in the DepMap mutation profiling dataset[43], further

supporting that YES1 baseline variation occurs and can be independent of genetic sequence. Together, this indicates the YES1 protein could be susceptible to a range of important conformational states or interaction partners that affect proteoform-level thermal stability between cell lines, and which could also limit or enable context-dependent ibrutinib binding.

Collectively, these examples illustrate how functional proteoform group analysis could enable a more nuanced interpretation of drug binding, one that is highly context and cell-line dependent. Additionally, comparing and contrasting results obtained at the aggregate gene symbol level and proteoform group level may indicate functionally relevant differences worth examining further.

## Validation by peptide resolved pulldown experiment

From qualitative proteoform identifications, some peptides may in fact be specific or overrepresented in specific proteoforms in a way that enables differential quantification. In these cases, to further validate the ability of our proteoform group-level drug binding approach to identify relevant drug-proteoform interactions, we performed an ibrutinib probe click chemistry pulldown using the RCH-ACV cell line and quantified proteoform groups by summation of PSMs. Results were considered significant if they were replicated in two out of three ibrutinib-treated preparations without DMSO detection, and results detected in both ibrutinib and DMSO pulldown samples were excluded from interpretation due to lack of statistical significance (Supplementary Data 5). The pulldown method has several notable limitations, namely that it detects only more stable, direct drug interactions rather than weaker interactions or those associated with protein complex effects, and this method is additionally limited by its high peptide detection threshold. Despite these considerations, we could nevertheless confirm 6 proteoform group-resolved pulldown hits from the 29 detected from peptide-matched RCH-ACV TPP proteoform hits, including multiple previously uncharacterized results (Fig. 4a; Supplementary Data 5). Among proteoform group results that were replicated in the pulldown, we identified the WASH complex component WASHC2C_1 (Fig. 4a, b), and both proteoform groups of BTK (Fig. 4c). We observed that WASHC2C_1 had higher mean pulldown intensity than the BTK proteoform group BTK_2 (WASHC2C_1 = $5.64 \times 10^6$, BTK_2 = $7.48 \times 10^5$), although it was only detected at the 20 μM ibrutinib probe dose. The second proteoform group that was not thermally impacted, WASHC2C_2, was also not detected in the pulldown (Fig. 4b). This preference of ibrutinib for WASHC2C_1 could hint at differential post-translational modifications, lipid interactions, or other structural differences consistent with a highly disordered protein (Fig. 4d) which might influence binding affinities between these proteoform groups. Alternatively, these differences might indicate that WASHC2C_1 is a more prevalent form in the RCH-ACV cell line, which also has implications for ibrutinib targeting. Taken together, our study indicated that MS-based proteomics coupled with TPP functional proteoform group detection can lead to comprehensive identification of unknown targets; more specifically, our results indicate that the relationship

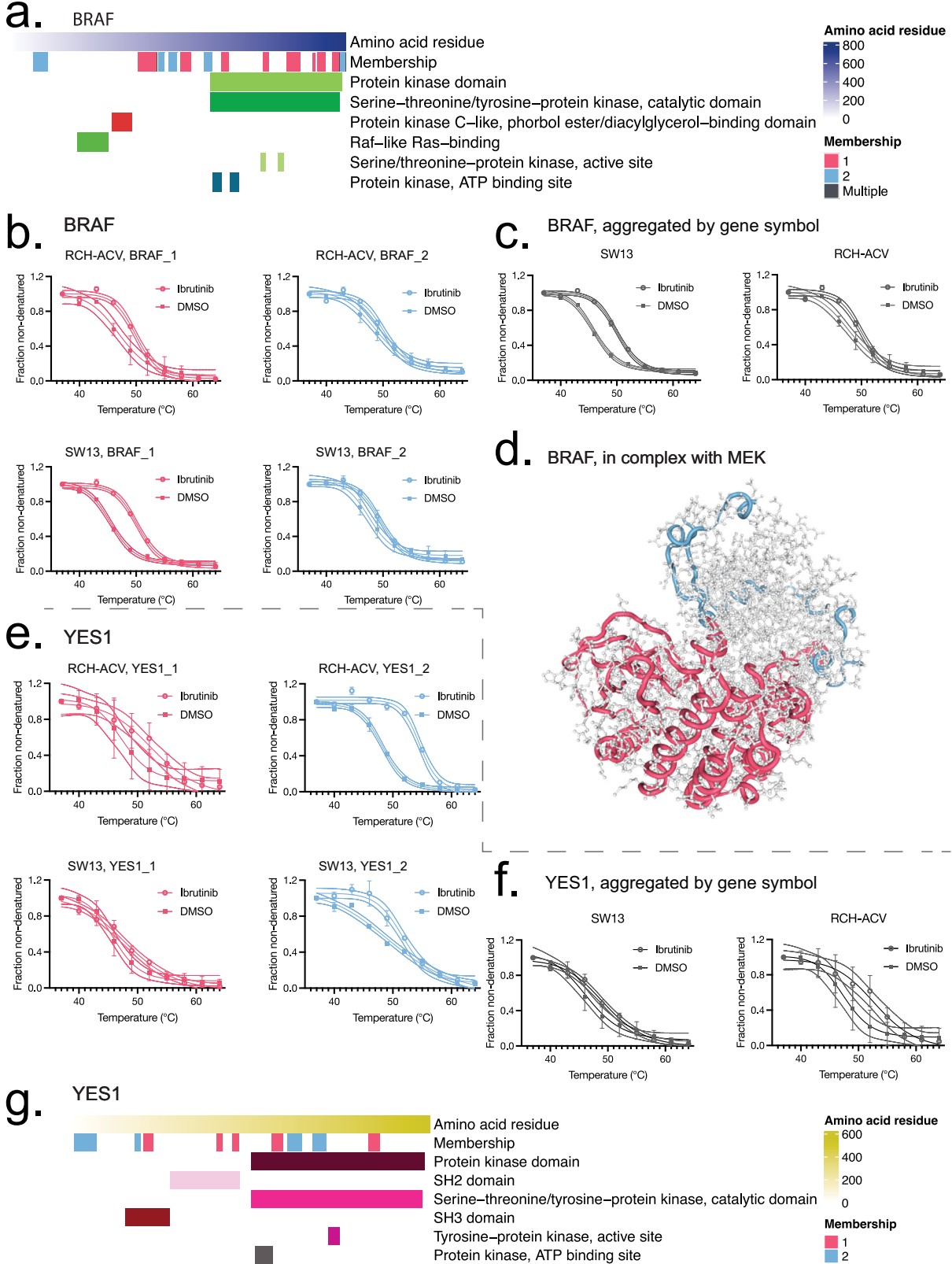

between the WASH complex and ibrutinib is likely physically robust and requires further study.

## Proteoform groups in CLL patients vary with treatment

A recent study generated comprehensive proteogenomics profiling of primary CLL samples and also quantified ex vivo sensitivity to ibrutinib in coculture[71]. Using this cohort of 68 patients profiled using HiRIEF LC-MS/MS proteomics, peptides were summed to functional proteoform groups and considered by their relative abundance, an independent metric not linked to their thermal behavior[27]. To account for technical aspects before interpreting the granularity lost or gained in the aggregation process, we performed f-tests with BH multiple testing

**Fig. 3 | Specific ibrutinib target detection enabled by functional proteoform group analysis. a** Peptides mapping to their position on the canonical FASTA sequence for BRAF (sp|P15056|BRAF_HUMAN), colored by functional proteoform group assignment (Membership) and highlighting interpro domain annotation. **b** Functional proteoform group melting behavior for BRAF, labeled by cell line, treatment, and proteoform. The melt curves for each condition represent $n = 20$ measurements from 10 temperatures for each experimental condition and cell line. Melt curves from 4PL curve fits show 95% confidence interval of the fitted model, each point shows the mean fraction non-denatured and error bars show ±SD. **c** Protein aggregated melting behavior for BRAF, separated by cell line and treatment. The melt curves for each condition represent $n = 20$ measurements from 10 temperatures for each experimental condition and cell line. Melt curves from 4PL curve fits show 95% confidence interval of the fitted model, each point shows the mean fraction non-denatured and error bars show ±SD. **d** Structural diagram of peptide mappings over the PDB structure for BRAF bound to MEK and an ATP analog, as described in ref. 68. Tube overlays color peptides by functional proteoform group assignments. Regions without assigned peptides or belonging to the MEK structure appear as a gray amino acid backbone. **e** Functional proteoform group melting behavior for YES1, separated by cell line, treatment, and proteoform. The melt curves for each condition represent $n = 20$ measurements from 10 temperatures for each experimental condition and cell line. Melt curves from 4PL curve fits show 95% confidence interval of the fitted model, each point shows the mean fraction non-denatured and error bars show ±SD. **f** Gene symbol aggregated melting curve for the YES1 protein, labeled by cell line and treatment. The melt curves for each condition represent $n = 20$ measurements from 10 temperatures for each experimental condition and cell line. Melt curves from 4PL curve fit show 95% confidence interval of the fitted model, each point shows the mean fraction non-denatured and error bars show ±SD. **g** Interpro domains and associated peptide mappings for YES1 functional proteoform groups. Source data are provided as a Source Data file.

correction for simulated functional proteoform groups assembled from randomized peptides within the gene symbol as a null distribution. This validated 2069 functional proteoform group hits as significantly variable, from among 10722 detected in the study, including many hits which were also thermally impacted by ibrutinib treatment (Fig. 5a). This supports our clustering assignment for these hits, which represent cases where the relevance of functional proteoform groups in treatment response biology is supported by evidence of comparative differences in abundance.

We next wondered whether off-targets were different in abundance for treated patients, potentially supporting functional hypotheses about indicated drug interactions. To perform this analysis, we assembled treatment data from patient history at time of sample for the cohort (Supplementary Data 6), identifying 3 were in ibrutinib treatment and 1 had been pretreated with ibrutinib. Proteoform groups were significantly altered between ibrutinib treated and untreated patients for 629 proteoform groups out of the 2069 with confident f-test variation differences (Fig. 5a, Supplementary Data 7), supporting that a portion of increased variability could be associated with treatment effects, despite the relatively low number of treated patients. These hits included thermally impacted hits from our TPP dataset, such as WASHC2C_1 (Fig. 5b), which was depleted, clarifying that the functional impact of the ibrutinib interaction lowers abundance and potentially limits function through direct antagonism or depletion. In contrast, WASHC2C_2 was not significantly impacted by ibrutinib treatment in our cohort (Fig. 5b).

Considering protein complexes enriched as off-targets, some demonstrated abundance changes, indicating their functional relevance. For the NUMAC complex, both proteoform groups of SCYL1 were significantly more abundant in ibrutinib-treated samples, and the proteoform group CARM1_2 was trending as downregulated, becoming significant (pAdj = 0.0380) when excluding the patient who had been pretreated and was not currently in ibrutinib treatment (Supplementary Fig. 11E). Given CARM1's pivotal role in chromatin remodeling and histone methylation[51], CARM1_2 downregulation could significantly impair the functions performed by the NUMAC complex. And upregulation of SCYL1 indicates adaptive mechanisms to maintain essential gene expression regulatory functions under pharmacological stress.

Additionally, the BRAF-MAP2K1-MAP2K2-YWHAE complex was supported by abundance changes for both BRAF and YWHAE, which were downregulated (Supplementary Fig. 11F). Intriguingly, BRAF_2 was the only significant downregulated proteoform group, although BRAF_1 was the proteoform group indicated in the TPP profiling and pulldown. Extending our previous BRAF proteoform group distinction hypothesis to a living cell, inhibitory ibrutinib interactions would not only be expected to target BRAF_1, but also limit undimerized BRAF incorporation into the RAF signaling axis as BRAF_2. Here, evidence of this functional impact is supported by the significant upregulation of

ARAF, demonstrating compensatory RAF signaling mechanisms[69] (Supplementary Fig. 11F).

For the COG complexes, the TPP data indicated many subcomplexes in a similar manner (Fig. 2a, Supplementary Fig. 8). But in the clinical cohort, COG proteoform groups were indicated much more specifically. Only COG6 was significantly altered for all proteoform groups, and while COG6_1 was highly downregulated (Fig. 5c), COG6_2 was upregulated. Among other significantly affected proteoform groups were COG1_1, COG7_1, COG8_1, and COG3_1, all of which were upregulated (Fig. 5c), and all of which also had at least one proteoform group in the dataset unchanged with treatment (Supplementary Data 7). This supports pharmacological interference limiting COG6_1, and functional compensation from other components. Considering the unique phenotype of glycosylation alterations noted in patients during ibrutinib treatment[57], these results cumulatively support that ibrutinib enacts highly specific reshaping of golgi function, and could inform a framework for future work better annotating COG complexes and resolving isoform and proteoform group-specific functions. Together, our findings accentuate the importance of dissecting proteoform-specific dynamics to unveil the broader spectrum of drug mechanisms and cellular adaptation.

## Divergent proteoform group profiles in a high splice patient subgroup with poor response to ibrutinib

In the study that first described the CLL cohort, a patient group was shown to have poor prognosis and rapid disease progression, termed ASB-CLL[71]. Generally, this patient group demonstrated reduced abundance and activity of components of B-cell receptor signaling, and enhanced abundance of components of the spliceosome. These samples included two patients undergoing ibrutinib treatment, but also many taken before treatment initiation. Overall, ASB-CLL was less responsive to ibrutinib in vitro in a coculture test (Wilcoxon, $p = 0.01$) (Fig. 6a), although still somewhat heterogeneous. Considering these results, it was evident that better delineation of proteoform groups relevant in ibrutinib treatment could improve the characterization of relevant components in ASB-CLL driving drug resistance.

Spearman correlations were generated for three subsets of the functional proteoform group abundance data: first, for all samples, and next for the ASB-CLL group and other samples separately. Among all samples, no correlations had especially high rho aligning with drug response, consistent with a heterogeneous population of samples. Only two results, SYNE2_1 and SMC3_1, achieved statistical significance both in correlation to drug resistance and in evaluating abundance variation relative to the null distribution; both are linked to nuclear architecture and cell cycle progression. And on the other side, only one functional proteoform group was linked to sensitivity, the RNA polymerase II transcriptional repressor LRRFIP1_1. Therefore, besides general indicators of cellular replication rates, stable biological differences linked to drug response were not well represented in a linear model of

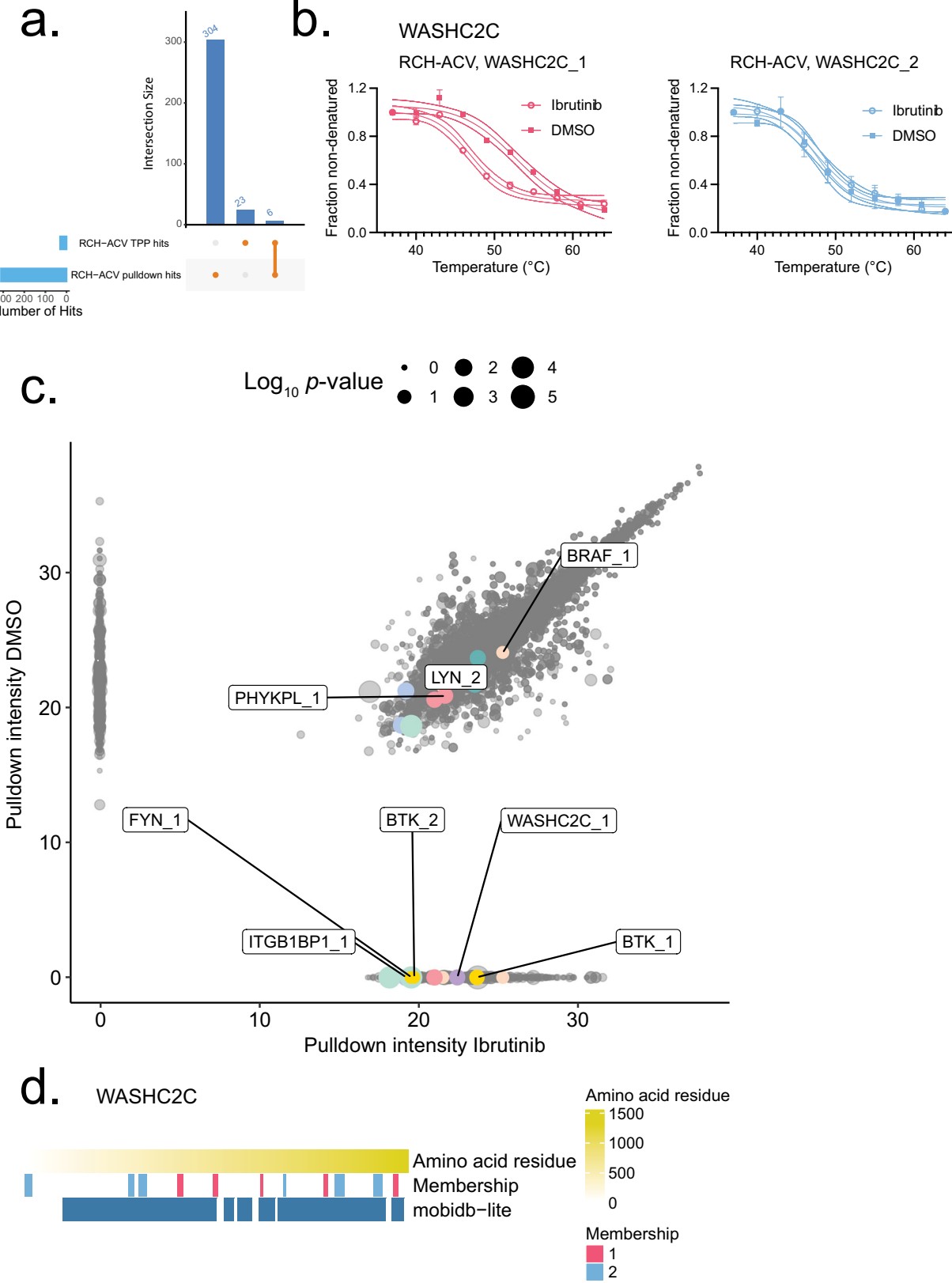

correlation to drug sensitivity among all samples, and drug resistance mechanisms in ASB-CLL are likely more specific to this subgroup rather than extensions of generalizable observations.

Considering ASB-CLL separately identified many high correlations towards both sensitivity and resistance, suggesting that their shared biological background could lead to different functional feedback loops and vulnerabilities in a treatment context. Notably, after filtering for significant abundance variation relative to the functional proteoform group null distribution, the best correlating result was also an ibrutinib binding target identified in the TPP data, BZW2_2 (Fig. 6b). This protein is known to have several isoforms and to regulate non-AUG initiated translation, and in the context of cancer, it is known to be

**Fig. 4 | Ibrutinib target validation by pulldown with functional proteoform group aggregation. a** Detected hits where results were in two preparations of ibrutinib probe without detection in DMSO controls, which were detected with peptides matching to functional proteoform groups in the pulldown data, showing overlap with RCH-ACV TPP data. **b** Functional proteoform group melting behavior for WASHC2C in RCH-ACV, separated and labeled by treatment and functional proteoform group. The melt curves for each condition represent $n = 10$ measurements each, each of the 10 temperature measurements performed in duplicates for each experimental condition. Melt curves are presented along with the 4PL curve fit and 95% confidence interval of the fitted model, each point shows the mean fraction non-denatured and error bars show ±SD. **c** Pulldown intensities for each replicate and dose, adjusted by $\log_2(\text{intensity} + 1)$. Point sizes indicate pAdj, corrected by BH, in the RCH-ACV TPP dataset, which were previously obtained using NPARC to distinguish melting in the other dataset. The pulldown was performed in triplicate experimental replicates for each condition. **d** Mobidblite mappings for WASHC2C peptides by functional proteoform group, representing disordered regions. Source data are provided as a Source Data file.

a MYC target[72,73] and to suppress MYC translation at its in-frame, non-AUG initiated isoform[74]. In the context of B- and T- cell receptor signaling, previous work has established that broad activation of protein translation is essential for sustaining signaling[75]. Intriguingly, links between BCR signaling, activation of translation, and efficacy of ibrutinib have already been well characterized in the context of CLL[76], where it was hypothesized that efficacy of ibrutinib may be in part linked to its inadvertent MYC modulation, in the sense that MYC translation is an essential oncogenic mechanism downstream of BCR signaling in CLL. Therefore, lack of the BZW2_2 functional proteoform group may demonstrate this previously characterized effect, representing lack of translational regulation and capacity for MYC oncogenesis independent of BCR signaling.

Despite this conceptual link to BCR signaling, BZW2_2 is unlikely to be physically associated with BCR components, and it was more prominently destabilized in SW13 where BTK and BCR kinases are not relevant to the lineage background (Fig. 6c). Additionally, other EIF2 family proteins were also destabilized (Fig. 6d) and indicated as components of the EIF2B2-EIF2B3-EIF2B4-EIF2B5 complex in the CORUM analysis within the SW13 background (Fig. 6d, Supplementary Data 3). Similarly to BZW2, EIF2 proteins are recruited to non-canonical start sites, but as translation initiators instead of suppressors[77]. Together, these results suggest a functionally relevant off-target axis for ibrutinib between similar proteins binding non-canonical translation start sites, likely to be dependent on underlying protein translation phenotypes and co-association in complexes, with potential lineage and cancer-specific effects.

## Discussion

This work demonstrates the potential of functional proteoform group deconvolution to identify new targets of drugs, using ibrutinib as a case study. Proteoforms can be inferred from untargeted thermal proteomics data and in relevant cellular contexts by applying our previous methods[27], and here we illustrate that functional proteoform groups can also be distinguished with respect to their drug binding abilities, leveraging the thermal impact of a drug for treatment-specific proteoform inference. This is supported by independent target indications in clinical CLL data, and these results enable deeper interpretation of the functional implications of drug activity and pave the way for identification and annotation of specific proteoforms and the roles they perform. We expect that functional proteoform group deconvolution will further expand the range of therapeutically relevant targets and improve the precision of personalized medicine. Moreover, using functional proteoform group analysis for target deconvolution could improve our understanding of adverse side effects, mechanisms of action, mechanisms of resistance, and polypharmacology.

Our study highlights the possibility that ibrutinib impacts many more off-targets than previously known, which may converge in their functional roles and impacted pathways. These results have far-reaching implications in interpreting the primary and secondary effects of ibrutinib treatment. For instance, ibrutinib-impacted functional proteoform groups described in our analysis play roles in mechanisms that may amplify drug efficacy, such as B-cell receptor signaling, induction of bone marrow egress, and T-cell

immunomodulation. We also uncovered other functions that were believed to be relevant clinically which may potentially be directly impacted by ibrutinib, such as Golgi trafficking, glycosylation, and cell adhesion. Collectively, the extended target list could enable elucidation of the complexities of BTK-independent ibrutinib immunomodulation.

Additionally, these results provide context for understanding and addressing multi-causal common effects of clinical importance. Aspergillosis is a very clinically common secondary infection[78] with unclear etiology to explain its high incidence, but which has been previously linked to endosomal defects[61,62] in addition to immunosuppression. Additionally, considering BRAF is activated by RAS-dependent dimerization[69], this mechanism leads to paradoxical transactivation of the pathway during V600E/K-specific BRAF inhibitor treatment, which has been identified to cause secondary skin cancers and general skin toxicities[79]. A severe toxicity in early ibrutinib trials was non-melanoma skin cancer, and recent meta-analyses have continued to reproduce the incidence of this as well as other cutaneous toxicities[80], which may be linked to proteoform-specific BRAF binding and paradoxical activation of the downstream pathway. More generally, our off-targets are consistent with receptor recycling defects, as previously identified for contributing to key clinically observed ibrutinib effects such as lymph node shrinkage and immune cell egress via modulating CXCR4[60]. Together, our results provide a foundation for examining and responding to mechanisms behind treatment outcomes.

Although this approach extends the capabilities of functional proteoform group-specific drug-target deconvolution, it has several limitations that are worth noting. In general, our functional proteoform group inference requires in-depth peptide detection, as well as significant instrument time and resources, which could potentially be improved by method optimization. Also, it is inherent to mass spectrometry proteomics that purely technical differences could change the identification and quantification of peptides, which will impact downstream clustering analysis and proteoform group assignment. Technical aspects are also especially important given the fragility of native proteins, where many factors are critical for recapitulating natural drug interactions, which leads to broad technical challenges in replication across many methods of drug-target deconvolution[81,82]. Although we opted to quantify complete melt curves, which enables a more cautious interpretation of technical aspects and regularity of melting behavior, alternative melting quantification approaches such as PISA[83] could support similar analysis while tailoring for high-throughput applications. Despite improving quantification of melt differences, focusing on full melt curve quantification limited our study's scope. Results from two cell lines indicated surprising biological variability, reducing statistical power, and the true biological variability between additional contexts may be higher. Future work could provide a more comprehensive understanding of variability by addressing inconsistencies when replicating experiments or comparing results across different studies. More specifically, although our experiments replicated a number of previously reported ibrutinib targets, we were not able to confirm all of them, most notably other TEC kinase family members and mutation-associated targets[84]. Therefore, tailoring additional experimental setups to capture more

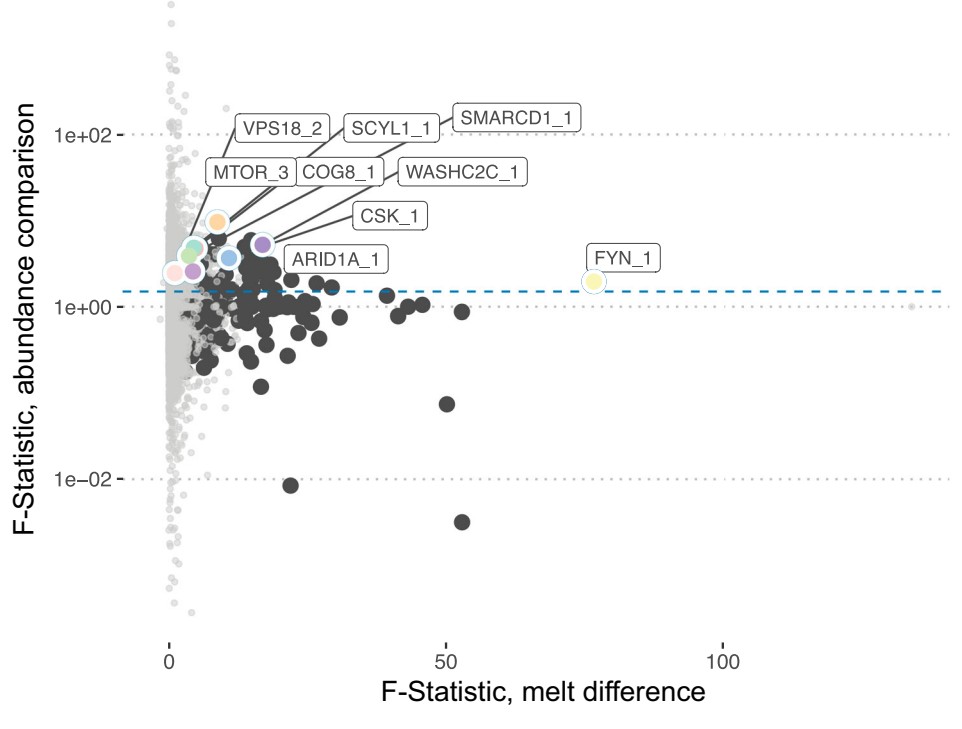

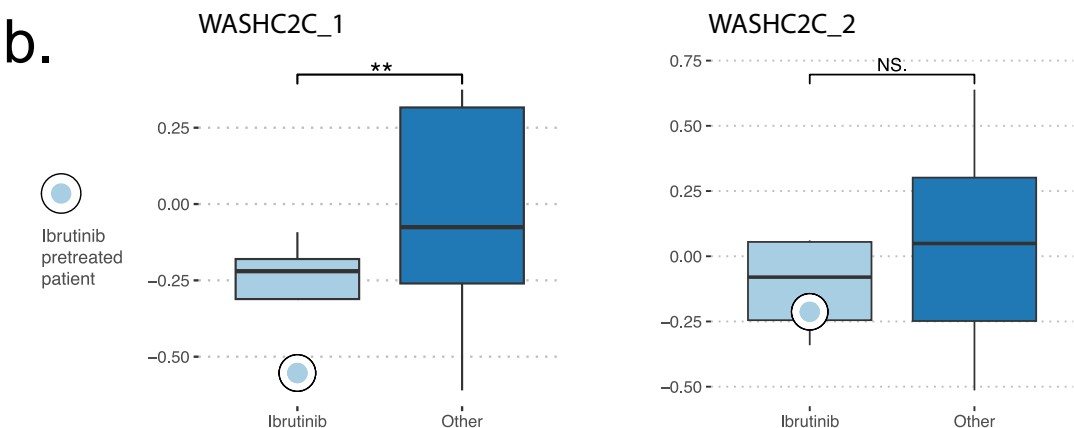

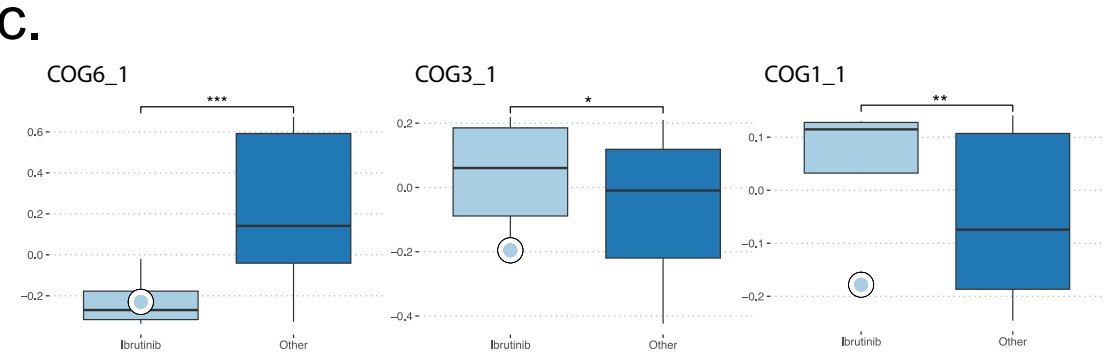

realistic cellular or tissue environments may be needed to address additional targets of drug activity. These wider ranging experimental settings could also promote interpretation of context-specific results more generally, such as by reconciling the status of the targets which were not detected in both cell lines in this study or which were not confirmed from other methodology. This may enable insight into

biological factors, or guide improvements in technical optimization. We hope that by introducing analytical frameworks for systematic delineation of functional proteoform groups, we might be able to achieve more unified results between methodologies, supporting future work scaling up this concept to have the technical power to explore systems-level questions.

**Fig. 5 | Ibrutinib treatment effects by functional proteoform group in a CLL cohort. a** F-tests were performed for variance in abundance in the cohort, compared against a null distribution of random re-subsetted membership identities within-gene symbol proteoform groups, and for melting differences in the RCH-ACV experiment. Dark points were significant results in this melting test, in SW13, or across all samples. Colored, labeled points were significant in both analyses. **b** WASHC2C_1 abundance in ibrutinib treated or pretreated samples compared to the rest of the cohort, evaluated using a two-sided Wilcoxon rank sum test with BH correction for multiple comparisons, pAdj = 0.00805. This effect was not detected for WASHC2C_2, pAdj = 0.172. This analysis used $n = 4$ biological replicate ibrutinib treated or pretreated patients with $n = 64$ biological replicate controls who

received another treatment or were not yet treated. The top and bottom edges of the box represent the first and third quartiles, with the median indicated by the line within each box. The whiskers extend to 1.5 times the interquartile range. **c** Selected COG complex components with abundance changes during treatment, evaluated using a two-sided Wilcoxon rank sum test with BH correction for multiple comparisons, COG6_1 (pAdj = $4.05 \times 10^{-5}$), COG3_1 (pAdj = 0.0446), and COG1_1 (pAdj = 0.00479). This analysis used $n = 4$ ibrutinib-treated or pretreated patients with $n = 64$ controls who received another treatment or were not yet treated. The top and bottom edges of the box represent the first and third quartiles, with the median indicated by the line within each box. The whiskers extend to 1.5 times the interquartile range. Source data are provided as a Source Data file.

Moving forward, we believe that our results and methods offer valuable insights for the refinement of preclinical research strategies and rationalize clinical observations. Continued research that extends these methods and incorporates additional therapeutic agents, cellular contexts, and functional interpretation approaches could enhance our understanding of drug mechanisms and lead to better-tailored precision medicine approaches.

## Methods

### Ethics

All procedures were conducted in accordance with the ethical guidelines and regulations approved by the Karolinska Institutet Research Ethics Committee and adhered to all relevant ethical standards for research. The collection of samples and clinical data, as previously published for the same cohort[71], was approved by the ethics commission of the medical faculty of the University of Cologne (13-091), the Department of Hematology Heidelberg (Ethics vote S-686/2018) and Etikprövningsmyndigheten (Ethical Review Authority) Dnr 2024-04186-01. Written consent was obtained from patients according to the declaration of Helsinki.

**Cell cultivation.** The childhood B-cell Precursor Acute Lymphoblastic Leukemia RCH-ACV (RRID: CVCL_1851) cell line was obtained from Deutsche Sammlung von Mikroorganismen und Zellkulturen GmbH (DSMZ, German Collection of Microorganisms and Cell Cultures, Braunschweig, Germany) and the adrenocortical carcinoma SW13 (RRID: CVCL_0542) cell line was obtained from American Type Culture Collection (ATCC). Roswell Park Memorial Institute (RPMI) 1640 (AQmedia, Sigma-Aldrich) supplemented with 10% fetal bovine serum (FBS, Sigma-Aldrich), 20 mM HEPES (Gibco/Life Technologies), 1 mM sodium pyruvate (Sigma-Aldrich), 1× MEM non-essential amino acids (Sigma-Aldrich), and 1x Penicillin-Streptomycin (Sigma-Aldrich) was used. Cell lines were grown at 37 °C and 5% $CO_2$ to a cell density of ~1–2 million cells/mL. Cells were harvested at $500 \times g$ for 3 min and washed twice with Hank's Balanced Salt Solution (Gibco™ HBSS, no calcium, no magnesium, no phenol red).

**Cell lysis and protein concentration.** RCH-ACV or SW13 cell pellets consisting of ~200 million cells were thawed on ice and resuspended in 3 mL of a HEPES (4-(2-hydroxyethyl)−1-piperazineethanesulfonic acid, Sigma-Aldrich) buffer (10 mM HEPES, 20 mM MgCl₂). Cells were subjected to three freeze-thaw cycles in liquid nitrogen and a 37 °C water bath, respectively, followed by mechanical disruption via syringe inversion. Protein concentration was determined using the DC protein assay (Bio-Rad) according to manufacturer-specified instructions.

**Lysate preparation and thermal proteome profiling.** Aliquots of 700 μL of RCH-ACV or SW13 cell extracts with ~2.3 mg protein per mL were treated with either 100 μM Ibrutinib or equivalent vehicle volume (DMSO) in duplicate experiments for ten min at 20 °C and with gentle shaking at 700 rpm. The treated lysates from each condition were aliquoted into ten 65 μL aliquots, and each duplicate experiment was

heated at 10 designated temperatures ranging from 37 to 67 °C in order to denature and aggregate the proteins. The remaining soluble protein fraction was cleared from the heat-aggregated proteins by means of centrifugation (40 min, $21,000 \times g$, 4 °C) and the resulting soluble protein fractions were prepared for liquid chromatography with tandem mass spectrometry (LC-MS/MS) analysis. Samples proceeded to digestion, desalting and mass-spectrometry proteomics data acquisition. First, the samples were diluted to contain 50 mM triethylammonium bicarbonate (TEAB, Sigma-Aldrich), 0.1% sodium dodecyl sulfate (SDS, Sigma-Aldrich) and 5 mM TCEP (tris(2-carboxyethyl)phosphine, Sigma-Aldrich). Reduction was performed at 65 °C for 30 min. The samples were then cooled down to room temperature and alkylated with 15 mM of chloroacetamide (CAA, Sigma-Aldrich) for 30 min. The proteins were digested overnight at 37 °C with a 1:70 Lys-C (Nordic Biolabs (Wako Chemicals GmbH)) to protein ratio and consecutively overnight at 37 °C with Trypsin (Thermo Fisher Scientific) at a 1:30 enzyme to protein ratio. The digested peptides were labeled using 10-plex tandem mass tag (TMT, Thermo Fisher Scientific), with labeled sets established using one 10-plex set for each corresponding 10 point melting curve and using the same amount of respective label for each sample. Labeling was performed according to the manufacturer's instructions but with 2-h incubation before quenching the TMT labeling reaction. Labeling efficiency was determined by LC−MS/MS before pooling the TMT-labeled samples.

**Desalting of peptides.** Desalting was performed using solid-phase extraction using SPE strata-X-C columns (Phenomenex). Prior to use, columns were conditioned by first wetting them with 1 mL of 100% acetonitrile (MeCN, Sigma-Aldrich), followed by a second wetting step using 1 mL of an 80% MeCN solution containing 0.5% formic acid (FA, Sigma-Aldrich). Next, columns were equilibrated by passing through 3 mL of 0.1% trifluoroacetic acid (TFA, Sigma-Aldrich). Subsequently, the samples were acidified to a pH range of 2–3 using FA before being loaded onto the cartridges. After sample loading, columns were washed and desalted by passing 3 mL of 0.1% TFA through them. This was followed by an additional wash with 0.25 μL of 0.5% FA. Peptides were then eluted from the columns using two 500 μL aliquots of 80% MeCN and 0.5% FA. Eluted peptides were subsequently dried using a SpeedVac (Thermo Fisher Scientific).

**High-resolution isoelectric focusing (HiRIEF) of peptides.** Prior to LC-MS/MS analysis, 300 μg of the TMT-tagged peptide pools were pre-fractionated using high-resolution isoelectric focusing (HiRIEF)[44]. Sample pools were subjected to peptide IEF-IPG (isoelectric focusing by immobilized pH gradient) in the pI range 3–10. Dried peptide samples were dissolved in 250 μL rehydration solution containing 8 M urea, and allowed to adsorb to the gel bridge strip by swelling overnight. The 24 cm linear-gradient IPG strips (GE Healthcare) were incubated overnight in an 8 M rehydration solution containing 1% IPG pharmalyte pH 3–10 (GE Healthcare). After focusing, the peptides were passively eluted into 72 contiguous fractions, first with MilliQ water, then with 35% MeCN, and lastly with 35% MeCN + 0.1% FA, using an

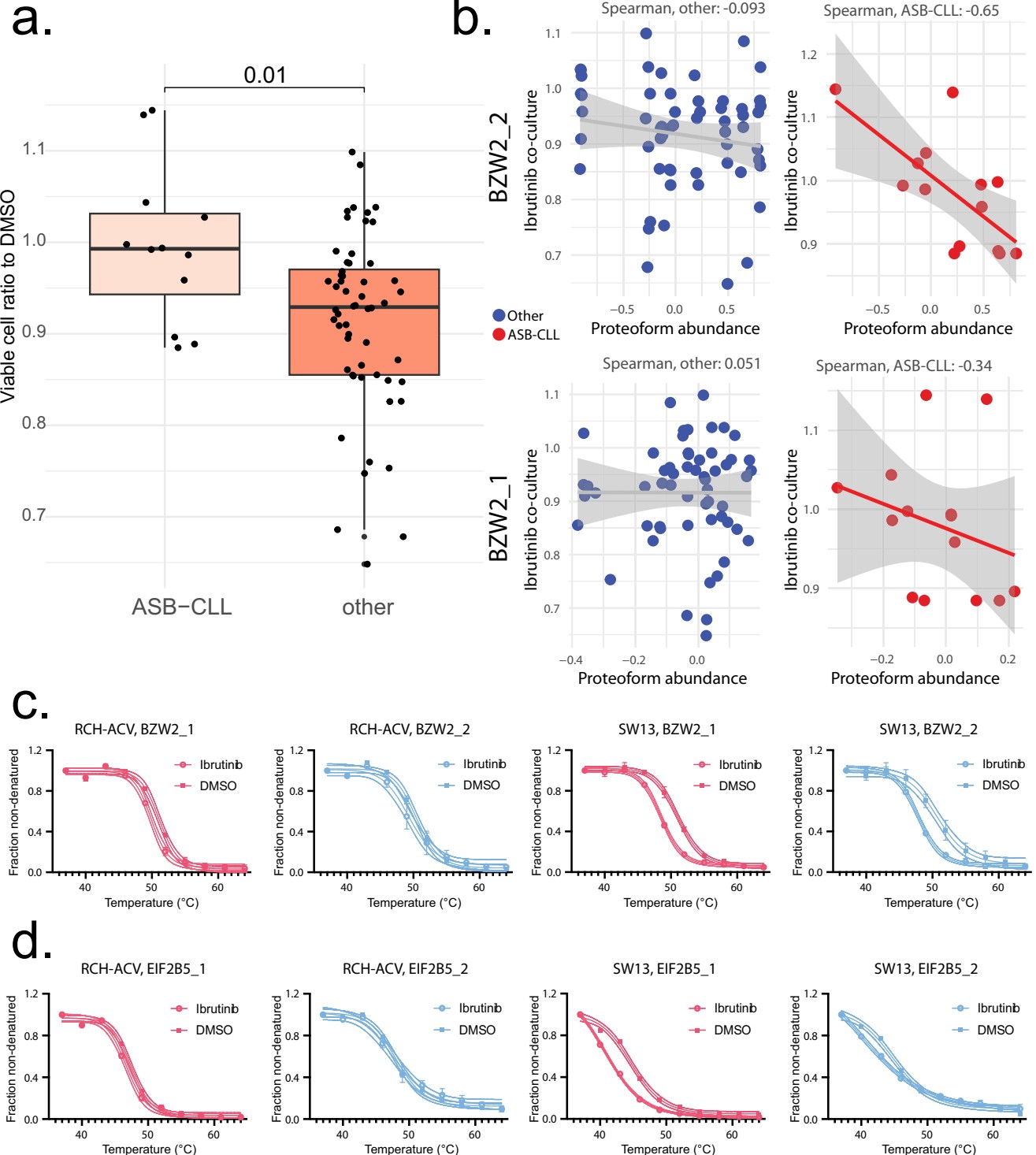

**Fig. 6 | Ibrutinib ex vivo sensitivity associations in ASB-CLL. a** Proportion of live cells in CLL samples, showing 40 nM ibrutinib response normalized to solvent control, obtained from ref. 71 and used as a baseline for correlation comparisons. This analysis used $n = 12$ biologically independent ASB-CLL samples compared against $n = 56$ biologically independent controls. A two sample Wilcoxon rank sum test is shown, $p = 0.01$. The top and bottom edges of the box represent the first and third quartiles, with the median indicated by the line within each box. The whiskers extend to 1.5 times the interquartile range. **b** BZW2_2 and BZW2_1 functional proteoform group abundances were correlated to coculture viability values. Spearman rho values and their 95% confidence interval (shaded gray area) are displayed in each plot. The $p$ value for the only significant result, between BZW2_2 and coculture response in ASB-CLL samples, was 0.0259, supporting that the −0.65 Spearman result is a plausible explanation rather than a null model where the correlation

is zero. **c** Functional proteoform group melting behavior for BZW2 proteoforms. The melt curves for each condition represent $n = 10$ measurements each, each of the 10 temperature measurements performed in duplicates for each experimental condition. Melt curves are presented along with the 4PL curve fit and 95% confidence interval of the fitted model, each point shows the mean fraction non-denatured and error bars show ±SD. **d** Functional proteoform group melting behavior for EIF2B5 proteoforms. The melt curves for each condition represent $n = 10$ measurements each, each of the 10 temperature measurements performed in duplicates for each experimental condition. Melt curves are presented along with the 4PL curve fit and 95% confidence interval of the fitted model, each point shows the mean fraction non-denatured and error bars show ±SD. Source data are provided as a Source Data file.

in-house constructed IPG extractor robotics device (GE Healthcare Bio-Sciences AB, prototype instrument) into a 96-well plate (V-bottom, Greiner product #651201). The resulting fractions were then dried in a SpeedVac and kept at −20 °C.

**LC-MS/MS runs of the HiRIEF fractions.** Online LC-MS/MS was performed using a Dionex UltiMate™ 3000 RSLCnano System coupled to a Q-Exactive-HF mass spectrometer (Thermo Fisher Scientific). Each fraction was subjected to MS analysis. Samples were trapped on a $C_{18}$ guard-desalting column (Acclaim PepMap 100, 75 µm × 2 cm, nanoViper, $C_{18}$, 5 µm, 100 Å), and separated on a 50 cm long C18 column (Easy spray PepMap RSLC, $C_{18}$, 2 µm, 100 Å, 75 µm × 50 cm).

Buffers used: nano capillary solvent A: 95% $H_2O$, 5% DMSO, 0.1% FA; solvent B: 5% $H_2O$, 5% DMSO, 95% MeCN, 0.1% FA.

At a constant flow of 0.25 µl min⁻¹, the curved gradient went from 2% B upto 40% B in each fraction, followed by a steep increase to 100% B in 5 min. FTMS master scans with 60,000 resolution and mass range of 300–1500 m/z were followed by data-dependent MS/MS scans (35,000 resolution) on the top 5 ions using higher energy collision dissociation at 30% normalized collision energy. Precursors were isolated with a 2 m/z window. Automatic gain control (AGC) targets were 1E6 for $MS^1$ and 1E5 for $MS^2$. Maximum injection times were 100 ms for $MS^1$ and 100 ms for $MS^2$. Dynamic exclusion was set to 30 s duration. Precursors with unassigned charge state or charge state 1 were excluded. An underfill ratio of 1% was used.

**Pulldown in lysate.** To ensure the inference of binding results is appropriate, an ibrutinib pulldown was performed using the RCH-ACV cell line, with results performed in triplicate and two probe concentrations. The lysates were treated with DMSO, 10 µM, and 2 mM of an ibrutinib-probe (PF-06658607, Sigma-Aldrich). For each treatment, samples were incubated for 10 minutes at 20 °C and with gentle shaking at 700 rpm. Afterward, lysates were centrifuged at 21,000 × g for 1-h at 4 °C. A premix containing Azide-Biotin (Jena Bioscience) (10 mM in DMSO), TCEP (52 mM, 15 mg/mL in ddH₂O), THPTA (tris-hydroxypropyltriazolylmethylamine, 1.667 mM in H₂O), and CuSO₄ (50 mM) was added to each sample. The samples were incubated at room temperature for 1-h. Post-incubation, the samples were treated with cold acetone for protein precipitation, stored overnight at −20 °C, and subsequently subjected to centrifugation at 21,000 × g for 15 min at 4 °C. Proteins were pelleted and washed twice with cold methanol. The pellets were resuspended using a probe sonicator and treated with 0.2% SDS in Dulbecco's phosphate buffered saline (DPBS, Gibco) with the addition of 0.6 M urea (Sigma-Aldrich). Protein concentration was determined, and equal amounts of protein were transferred to Protein LoBind tubes containing prewashed (1 mL 0.2% SDS in DPBS) neutravidin beads (Sigma-Aldrich). Samples were incubated for 1-h under continuous mixing, followed by washing with 0.2 % SDS in DPBS, 6 M urea in ddH₂O, and DPBS. Following this step, samples proceeded to on-bead digestion, desalting and mass-spectrometry proteomics data acquisition. The data search was performed as described below. Results were considered significant if they were replicated in at minimum two out of three ibrutinib treated preparations without DMSO detection. Results detected in both ibrutinib and DMSO pulldown samples are not discussed or included in the hits count, but are included in the supplementary data 5 if they were replicated in at least two preparations.

**On-bead digestion.** The beads were resuspended in a buffer containing 2 M urea in 50 mM TEAB and samples were subjected to reduction and alkylation steps with 50 mM TEAB, 0.1% SDS, and 5 mM TCEP. 1:40 Lys-C enzyme (Nordic Biolabs (Wako Chemicals GmbH)) was added, and samples were incubated overnight at room temperature. Trypsin digestion was performed at a 1:70 enzyme-to-protein ratio for an 8-h incubation at 37 °C.

**LC-MS/MS runs of the pull-down.** Online LC-MS was performed using a Dionex UltiMate™ 3000 RSLCnano System coupled to a Q-Exactive-HF mass spectrometer (Thermo Fisher Scientific). Samples were trapped on a $C_{18}$ guard-desalting column (Acclaim PepMap 100, 75 µm × 2 cm, nanoViper, $C_{18}$, 5 µm, 100 Å), and separated on a 50 cm long $C_{18}$ column (Easy spray PepMap RSLC, $C_{18}$, 2 µm, 100 Å, 75 µm × 50 cm). A 3-h gradient was run with the following gradient profile: 0-6 min: 3% 12 min: 6% 185 min: 37% 190 min: 42% 192 min: 99% 200 min: 99% 203 min: 3% 213 min: 3%.

Detection Settings: First scan: 70,000 resolution, 1E6 AGC target, 100 ms max IT, mass range 300–1600 m/z. Data-dependent MS/MS: 35,000 resolution, 1E5 AGC target, 150 ms max IT, top 5 ion selection, 2 m/z isolation window, 1E3 min AGC target, 30 s dynamic exclusion.

Buffers Used: NanoA: 95% $H_2O$, 5% DMSO, 0.1% FA, NanoB: 90% MeCN, 5% DMSO, 5% $H_2O$, 0.1% FA LoadA: 97% $H_2O$, 3% MeCN, 0.1% FA LoadB: 95% MeCN, 5% $H_2O$, 0.1% FA.

**Mass spectrometry data search.** Raw mass spec outputs were processed for quality control and quantified with a standardized pipeline, ddamsproteomics version 1.0.2, openly accessible at: https://github.com/lehtiolab/ddamsproteomics/releases/tag/v1.0.2.

The following adjustable parameters were specified: --genes --hirief --fractions --symbols --isobaric tmt10plex --denoms 'DMSO1:126 DMSO2:126 IBRUTINIB2:126 IBRUTINIB1:126'.

The mapping database was Homo_sapiens.GRCh38.92.

**Functional proteoform group identification.** Quantitative reporter ion signals for PSMs were summarized to peptides by summation. Reporter ion signals of all individual temperatures were normalized using variance stabilizing normalization and converted to fold changes relative to the first temperature. Next, a graph for each gene symbol was created connecting all peptides (vertices) with weights (edges) corresponding to their similarity in melting profile, to assign similar melting peptides. The similarity was computed using weighted Euclidean distance, according to the formulas as described[27]. Obtained graphs were then used for community detection using the Leiden algorithm. Only gene symbols for which at least ten peptides were identified and with at least two peptides per sample were used as input for graphs (a detected community had to be supported by at least three peptides to be accepted to ensure that outlier peptides did not affect robust functional proteoform group identification). Peptides mapping to gene symbols for which these criteria were not fulfilled were grouped to single proteoform groups, and peptides mapping to gene symbols that were included in the community detection were assigned to proteoform groups if the modularity of the detected communities was higher than $1 × 10^{-13}$ and the peptide ambiguity ratio was lower than 0.5 (for peptides mapping to multiple genes, it is calculated as the number of ambiguous peptides divided by the sum of the number of gene-specific and ambiguous peptides). Modularity was computed using the function modularity() of the igraph R package. Through the assignment of peptides to communities, functional proteoform groups for each gene symbol were created. Summarization to proteoform groups was performed by summation of non-normalized raw peptide data assigned to individual communities. Obtained functional proteoform group signal intensities were then normalized per temperature using variance stabilizing normalization, and relative fold changes to the lowest measured temperature were formed.

Differential melting curve analysis was performed using NPARC, as previously published and described in the context of this analysis[27,45]. NPARC compares protein stability between treatment groups by fitting two competing models to the melting curve data: a null model that assumes no difference between conditions and an alternative model that allows for condition-specific differences in protein thermal stability. Sigmoidal melting curves are fitted to the data using smooth functions. An F-statistic is calculated to quantify the

relative improvement of the alternative model over the null model. *P*-values are computed based on an approximate F-distribution with effective degrees of freedom estimated from the data, and multiple testing correction is performed using the Benjamini-Hochberg method. Each comparison was filtered to consider only full duplicated melting curves in each treatment group.

**Sequence composition profiling.** Canonical amino acid sequences were fetched from the Swissprot assembly on Uniprot using the Rcurl package (5.2.0) (see: https://github.com/isabelle-leo/ibrutinib_proteoforms, the github repository for the study). Sequences for each matched gene symbol were evaluated as ibrutinib targets using NPARC[27,45], and were included if they were indicated by any or all lineage NPARC test with $p < 0.05$. The proportion of sequences was compared to the proportion in the detection background for the full experiment. Proportional tests using 2-sample test for equality of proportions with Yates' continuity correction were performed according to ref. 85.

**Over-representation and network topology analysis.** Over-Representation Analysis (ORA) and Network Topology-based Analysis (NTA) were performed using the R package WebGestaltR, version 0.4.6. The NTA was executed utilizing the WebGestaltR function, and Biological General Repository for Interaction Datasets (Biogrid) analysis was obtained utilizing the Homo Sapiens network_PPI_BIOGRID enrichment database, as included in the package. Gene symbols with an adjusted *p*-value (pAdj) below 0.05 were used as the genes of interest, with pAdj obtained from NPARC analysis of all melt curves and including only results with eigth full melting curves. The resultant network was processed with the Network_Retrieval_Prioritization method. Data was output for the top 100 results. Protein complex enrichment was performed using ORA with the Comprehensive Resource of Mammalian Protein Complexes (CORUM) database, also retrieved in the R package as the Homo Sapiens network_CORUM enrichment database. The full list of unique gene symbols that had 8 full melting curves were used as reference genes, and gene symbols with significant thermal melting changes as identified with NPARC were used as the hits input. Significance was adjusted (pAdj) using the FDR method Benjamini-Hochberg, BH, with a threshold of 0.05. A minimum number of three gene symbols in each CORUM complex category was required when assembling the input database. Multiples of gene symbols representing different functional proteoform identities were considered separately, in both the enrichment list and the background list, enabling the analysis to consider the background chance of a gene symbol's random identification accurately, and to prioritize cases where proteoform groups under the same gene symbol category share meaningful engagement across multiple measured protein contexts.

**CLL clinical data processing.** The dataset includes 68 independent patient samples, and 72 total samples including replicates, outlined in supplementary data 6. The HiRIEF data was obtained from PRIDE, accession: PXD028936 [https://www.ebi.ac.uk/pride/archive/projects/PXD028936]. The PSM values were VSN normalized by set. Then, the PSMs were summed by the following three parameters: by gene symbol, by functional proteoform group (for matched peptides), and by random functional proteoform group (shuffling only the membership identification, also for matched peptides). When selecting random peptides, the functional proteoform group membership identifier was scrambled for the same set peptides, maintaining the proportion of matched peptides per gene symbol to be comparable as possible to the functional proteoform group summed data. After summation, the data was log-transformed and median centered. The random peptide data set was used as a null model in f-tests to evaluate functional proteoform group variation and underlying noise in peptide-level data.

## Reporting summary
Further information on research design is available in the Nature Portfolio Reporting Summary linked to this article.

## Data availability
The mass spectrometry proteomics data and processed clustering analysis data have been deposited to the ProteomeXchange Consortium via the PRIDE partner repository with the dataset identifier PXD047187. Annotations of proteins were based on the Ensembl 92, GRCh38.p13 human genome assembly released in April 2018. Source data are provided in this paper as Source Data Files. Source data are provided with this paper.

## Code availability
All code used to perform the computational analyses described and to reproduce the figures is hosted at: https://github.com/isabelle-leo/ibrutinib_proteoforms. This code may be updated at a future date. The code has also been deposited in zenodo: https://doi.org/10.5281/zenodo.14134715. And the pipeline used to perform the DDA-MS search is available on both github, https://github.com/lehtiolab/ddamsproteomics/releases/tag/v1.0.2, and zenodo, https://doi.org/10.5281/zenodo.14134606.

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

## Acknowledgements
This study was supported by grants from the Swedish Cancer Society (23 2843 Pj and 24 0866 SIA); Swedish Childhood Cancer Foundation (PR2022-0009); the Swedish Research Council (R.J., 2017-01653); the Dr. Åke Olsson Foundation for Hematological Research (2021-00130); and Cancer Society Stockholm and the King Gustaf V Jubilee Fund (R.J., 204092 and 221132); R.J. acknowledges the Karolinska Institutet and Science for Life Laboratory. For contributions in executing and maintaining the mass spectrometry proteomics search pipeline, the authors would like to acknowledge Jorrit Boekel. The authors would also like to thank Sascha Dietrich, Matthias Stahl, Luay Aswad, Xuekang Qi and Henri Colyn Bwanika for their valuable input.

## Author contributions
R.J. designed and coordinated the study, and provided funding. J.L. provided MS and analysis platform together with R.J. I.R.L. and E.K. developed the methodology and performed mass spectrometry experiments with help from A.A. and J.E. E.K., A.A., and M.T. performed the TPP experiments. E.K. and J.E. performed the pulldown experiments. I.R.L. developed data analysis strategies and analyzed the data. I.R.L. drafted the manuscript with input from R.J. which was reviewed and edited by all authors.

## Funding

## Competing interests
All authors declare that they have no competing interests. E.K. is a current employee at Evotec, with the position beginning after contributions to this work concluded.
