## [Transparent Peer Review file · Nature Communications]

Functional proteoform group deconvolution reveals a broader spectrum of ibrutinib off-targets

Corresponding Author: Dr Rozbeh Jafari

A version of this paper was originally rejected for publication by Nature Communications, however that decision was reconsidered after appeal by the authors.

Version 1:

Reviewer comments:

Reviewer #1

(Remarks to the Author)

In their manuscript "Proteoform-level deconvolution reveals a broader spectrum of ibrutinib off-targets", Isabelle Leo and co-authors developed a proteoform specific TPP approach and applied it to a detailed screen for the identification of ibrutinib off-targets. The authors successfully demonstrate the power of their new approach by identifying and validating novel off-targets that could not be found by previous methods. The core novelty of the approach is certainly the extraction and quantification of specific proteoforms from the TPP data. While the authors describe this in detail, the corresponding data is not shown and should be included as supplemental material to better understand and reproduce these crucial steps. Overall, the authors describe an interesting workflow with great potential to enhance the functional information obtained by the popular TPP technique. The manuscript is well written and the data clearly presented. The conclusions are also supported by the data. I have only a few minor comments that need to be addressed before publication.

Specific Comments:

Supplemental Table 1: In the opinion of this reviewer including some information about the different Proteoforms, ideally with some common annotation like UniProt, would be helpful to better understand the table and the found hits in general. Along these lines, the authors did use a very sophisticated workflow for assigning peptides to different proteoforms and the identification thereof. This proteoform assembly would be highly useful information for the readers and should be provided as a supplemental table listing all identified proteoforms with their corresponding peptides.

Figure 1C: This figure is a bit hard to read. The label "DMSO" and "IBRUTINIB" is present 2 times. To keep it consistent with the supplemental Figure 1B, this reviewer would suggest to remove the legend and just keep the headers, maybe adding the protein name as done in all other plots of this kind. Furthermore, the authors most likely wanted to show that the two proteoforms of BTK have different melting curves, which is fine, but it would help for clarity to state this clearly in the text.

Figure 3B, C, E, F, 4C: Again, here the "membership" 1 and 2 is also represented directly in the header of each plot. This might be confusing. Please just use the header which is the most obvious information readers will look at.

Page 19, line 440: The authors rightly mention that TPP requires a substantial amount of instrument time for in-depth analysis. Would the strategy presented involving proteoforms also work with the more straightforward and faster new PISA approach (10.1021/acs.jproteome.9b00500) in which the different temperature fractions are combined before LC-MS analysis? Here, instead of 10 temperature fractions per sample, only one sample has to be analysed.

(Remarks on code availability)

I was not able to access and download the code. The authors state that a copy is provided in the submission, but this was not found either.

Reviewer #2

(Remarks to the Author)

Leo et al. present results of TPP experiments attempting to analyse ibuprofen off-targets by exploiting proteoform-level deconvolution. The paper is nice contribute to the literature on using proteoform level analysis to gain deeper insights. Whilst I'm generally supportive of the manuscript it felt somewhat rushed in places and perhaps a bit preliminary. I have several comments which I think will help improve the manuscript.

1) To address my comments around feeling rushed, please look through your figures carefully Fig. 2B has no y-axis ticks, in this figure the sizing is also different and style changes throughout. Harmonising this will make it much easier to follow. Fig. 3 the figure are out of order and generally hard to follow logically. Fig. 5 has many axis labels missing and general spacing could be improved. Fig. 6 again sizing is an issue and panel A is confusing the y-label is labelled "value" which isn't very helpful and if this is a proportion (which is how I read the legend) then why are the values over 1?

2) The communication of the statistics is very confusing. Why do the author keep presenting non-adjusted results alongside adjusted results, surely the non-adjusted hits are just full of false positives?

3) Continuing with the statistics, I feel a careful control is missing here. Because of the clustering that happens earlier, it can be easy to overstate the statistical significance. It would be useful to see an important negative control where your compound doesn't affect any proteins. Applying the same pipeline the authors should see no proteoform level changes. The way that the analysis is currently done I'm unsure if this would be the case.

4) Replication is not always clear, it useful to see explicitly in figures what the number of biological vs technical replicates was.

5) It would be useful to see these results alongside results without proteoform level deconvolution (perhaps this is there but maybe not signposted well?). Then it would be easier to see the benefits of this proteoform level deconvolution.

6) The prior work on this topic seems a little sparse, the authors should make clear they are applying the methodology of their previous paper. It doesn't read so generously in this current paper.

7) The follow is a bit preliminary. I would be keen to see an assay on a protein where ibuprofen supposedly binds to one proteoform but not the other. Can you express both proteoforms and confirm this?

(Remarks on code availability)

Reviewer #3

(Remarks to the Author)

In their manuscript "Proteoform-level deconvolution reveals a broader spectrum of ibuprofen off-targets", Leo et al., describe the application of their recently developed method (Kurzawa et al, Nat. Chem. Biol., 2023) to study the effects of ibuprofen on CLL cells at "proteoform-level" using thermal proteome profiling. Their method represents exciting developments in the field and I believe their new application could be an important step to promote future developments in this direction. However, there are several critical components that need to be revised.

Major issues:

1. Terminology

The term "proteoform" has been narrowly defined to represent the exact combination of amino acid sequence and post-translational modifications (Smith & Kelleher et al., Nat Methods, 2013). Within the top-down proteomics community, which measures intact proteins, this has become an established term.

The term is also used within bottom-up proteomics, which measures mixtures of digested proteins, but typically only in introductions and discussions: Intrinsically, bottom-up proteomics can not measure proteins or proteoforms directly, but only conduct inference based on the identified peptides. This already poses a problem for the identification of canonical proteins when only shared proteins are present. In this case, the term "protein group" is used to attribute for the ambiguity.

I was confused that the authors decided to use the term "proteoform" throughout their manuscript, as it is impossible to ensure by bottom-up proteomics that the identified peptides correspond to 1 to N specific proteoforms. I found it even more confusing that in their original method paper (Kurzawa et al, Nat. Chem. Biol., 2023), they proposed a very suitable terminology themselves, calling those events "inferred functional proteoform groups". Since this subfield associating function to proteoform groups is still very young, I believe it is critical to establish very clean terminology.

1.1: The authors should ensure that their originally proposed term "proteoform group" is used throughout the manuscript were applicable. The term "proteoform" should never be used when results could be ambiguous.

1.2: In the introduction, the authors should briefly establish the definition and relationship between "proteoform", "proteoform group" and the challenges related to classical protein inference and grouping in bottom-up proteomics.

2. Proteoform identification in ibuprofen treated cell lysates

2.1: "Although BTK_1 was the only BTK proteoform that met the $p_{Adj} < 0.05$ NPARC test significance threshold, BTK_2 was significantly shifted based on a $p < 0.05$ threshold."

This statement implies statistical significance for BTK_2, however, only if multiple-testing correction is ignored. This is not appropriate and instead, the authors should report the multiple-testing corrected value for BTK_2 as well. They can still do all comparisons and discussions downstream even if BTK_2 is not a significant hit.

2.2: "We used BTK results to calibrate significance levels for further result interpretation, and we proceeded to consider $p_{Adj} < 0.05$ as likely thermally impacted, and $p < 0.05$ as plausibly thermally impacted."

Multiple-testing corrected and uncorrected thresholds should never be mixed. Instead the authors should report the corresponding more liberal multiple-testing corrected p-value for plausibly thermally impacted hits.

2.3: "In addition to BTK, previous studies¹⁸ have revealed that ibrutinib binds a wide range of proteins (Supplementary Table 2), and our analysis confirmed several of these (Supplementary Figure 1C)."

Supplementary Figure 1c indicates that the overlap of the SW13 and the RCH-ACH TPP datasets is 6.8%. This seems extremely low and could be a result of multiple confounding factors, e.g. a) extreme biological diversity of cell lines, b) concentration-cell-dependent response to ibrutinib, c) high specificity of the method, but low sensitivity, d) poor technical reproducibility, etc.

The authors should put this low overlap into context and try to explain factors contributing to it.

2.4: Ibrutinib concentration: "RCH-ACV or SW13 cell extracts were treated with either 100 μ M ibrutinib or equivalent vehicle volume (DMSO) in duplicates for 10 min at 20°C and with gentle shaking at 700 rpm."

How was the applied concentration of ibrutinib selected? Ibrutinib seems to have a very low IC₅₀ of <0.5nM (PMC3751776). What happens if a concentration 200,000-times higher than IC₅₀ is applied? I'm not sure if I'm missing a point here, but the authors should provide a clear rationale for how the drug concentration was selected, as this can have a tremendous effect on the context of the results and invalidate the whole concept of a study. When screening drug compounds in vivo, ideally, selected concentrations should be within a physiological scale similar to effective drug concentrations in tissue. I understand that for this study, screening was not conducted in vitro but in vivo. But such a dramatic difference in drug concentration will have a tremendous effect on off-target detection, which might be entirely absent or irrelevant in clinical settings.

The authors should demonstrate and discuss whether their results can translate to in vivo or clinical scenarios, and if not, what limitations (e.g. context-specificity of observed off-target effects) should be expected.

3. Using proteoform data to detect effects of drug binding on protein-protein interactions and complex formation

3.1: "Protein-protein interactions are highly dependent on cell lineage⁵³, and we observed that the baseline thermal stabilities of complex-associated proteoforms and magnitude of drug-induced thermal changes were not uniform between cell lines."

I think this is an extremely critical and potentially impactful part of the study. However, I'm worried about the comparison and interpretation of the results. If I understood correctly, differential analysis was conducted of the two cell lines separately of ibrutinib vs DMSO. Then, the two lists of significant hits were compared to assess potential differences. Unfortunately this will exclude hits with sparse coverage across all samples and call those differential, which are at the border of significance, right?

One approach for a more statistically meaningful comparison would be to employ an "interaction model" (e.g. as implemented in LIMMA) to compare (SW13_ibrutinib / SW13_DMSO) / (RCH-ACV_ibrutinib / RCH-ACV_DMSO). The authors should reformulate their differential comparison to ensure that differentially identified interactions are really significant and not observed due to technical artefacts.

4. Validation by peptide resolved pulldown experiment

4.1: "Results were considered significant if they were replicated in two out of three ibrutinib treated preparations without DMSO detection, and results detected in both ibrutinib and DMSO pulldown samples were considered if they were replicated in at least two preparations and also had intensities that met a paired t-test threshold of $p < 0.05$ "

The authors should provide multiple-testing corrected p-values and thresholds for pull downs as well. It is acceptable if this results in a large cutoff, e.g. 20-50% FDR, but it is critical that readers have the opportunity to judge the confidence of the results.

5. Proteoforms in CLL patients vary with treatment

5.1: "Using this cohort of 68 patients profiled using HiRIEF LC/MS proteomics, peptides were summed to proteoforms and considered by their relative abundance, an independent metric not linked to their thermal behavior²⁷."

This part would benefit from more explanations. To make sure that I understand this correctly: Across the 68 CLL patient

cohort, an F-test was conducted testing the intensities of those peptides grouped to a functional proteoform group by TPP, against the intensities of a randomized selection of peptides mapping to the same gene identifier measured in the 68 CLL patient cohort.

Would we not expect the randomized selection to have intrinsic bias, as they could not uniquely associated to a functional proteoform group by TPP? Presumably, those could be shared peptides that intrinsically have higher abundances?

Also, I struggle with the quantitative comparison of different peptides: Quantitative protein inference algorithms such as MaxLFQ work because they can operate on shared sets of peptides. Here, the comparisons are only conducted on sets of unique peptides. An appropriate comparison would be to first normalize raw peptide intensities against matching healthy tissue, which is provided by most label-based studies. Maybe the authors have done this, but from their methods section, this was not clear, as they write "The PSM values were VSN normalized by set. Then, the PSMs were summed by the following 3 parameters", which suggests label-free spectral counting. In absence of matching healthy tissue samples, the authors could redo the analysis with any non-CLL samples that have similar proteome coverage to demonstrate that indeed proteoform groups drive differential abundance and not the selection of peptides.

The authors should clarify the methodological assumptions and demonstrate that peptide intensity and not peptide selection is driving differential proteoform group abundance.

Minor issues:

- All figures: Please indicate statistical significance on all melting curves.

- Fig 1a: I don't understand the rationale behind Fig. 1a. Please either extend the description to guide the reader or better, replace this figure with a schematic of the experimental design.

- Fig 1d: This figure is way too small.

- Fig 1e: This is an extremely non-standard figure. Is this just a barplot to display significance? This could be omitted or illustrated in a much easier way.

(Remarks on code availability)

The R-code is provided as an archive including a brief README file.

Version 2:

Reviewer comments:

Reviewer #1

(Remarks to the Author)

The authors have addressed all my comments satisfactorily.

I have also been asked to comment on the concerns of reviewer #3.

Reviewer #3 raised valid technical concerns regarding the use of the term proteoform, statistical data analysis and presentation of the data as well as the use of a very high drug concentration in the experiments. In my view, the authors have very carefully and convincingly addressed all comments raised by Reviewer #3, also by adding new results. Thus, in my opinion, the manuscript is now considerably improved and in an appropriate state for publication.

(Remarks on code availability)

Reviewer #2

(Remarks to the Author)

The manuscript is greatly improved in response to the revision and appears to be acceptable. I don't need to see a revised version but I note there is heavy reliance on technical duplicates, which on one hand results in reduced power and on the other hand could underestimate biological variability. I appreciate from a time and cost perspective this may be the only option but the limitation should at least be mentioned in the discussion, so those doing similar experiments can understand why their might be inconsistencies.

(Remarks on code availability)

Nature Communications manuscript NCOMMS-24-01474A-Z Rebuttal

REVIEWER COMMENTS

Reviewer #1 (Remarks to the Author):

In their manuscript “Proteoform-level deconvolution reveals a broader spectrum of ibrutinib off-targets”, Isabelle Leo and co-authors developed a proteoform specific TPP approach and applied it to a detailed screen for the identification of ibrutinib off-targets. The authors successfully demonstrate the power of their new approach by identifying and validating novel off-targets that could not be found by previous methods. The core novelty of the approach is certainly the extraction and quantification of specific proteoforms from the TPP data. While the authors describe this in detail, the corresponding data is not shown and should be included as supplemental material to better understand and reproduce these crucial steps. Overall, the authors describe an interesting workflow with great potential to enhance the functional information obtained by the popular TPP technique. The manuscript is well written and the data clearly presented. The conclusions are also supported by the data. I have only a few minor comments that need to be addressed before publication.

Thank you for the feedback and positive remarks on the manuscript. The code was intended to be fully accessible during review, and it's unfortunate that this wasn't available to the reviewer. While the data and code are sufficient to fully reproduce the generation of peptide clustering results, we certainly appreciate the added value of making our results more accessible by providing the intermediate peptide membership data. This has now been added to the raw data repository. In line with other reviewer comments, we have also taken specific steps to improve the wording and figures.

Specific Comments:

Supplemental Table 1: In the opinion of this reviewer including some information about the different Proteoforms, ideally with some common annotation like UniProt, would be helpful to better understand the table and the found hits in general. Along these lines, the authors did use a very sophisticated workflow for assigning peptides to different proteoforms and the identification thereof. This proteoform assembly would be highly useful information for the readers and should be provided as a supplemental table listing all identified proteoforms with their corresponding peptides.

Thank you for this suggestion, we agree that this will improve accessibility and should be added to our data repository.

Action taken: We have uploaded an R data file with the full clustering intermediate analysis object, for all peptides in the data set, which can be easily used alone or as a time saving input for other analysis in the paper.

Figure 1C: This figure is a bit hard to read. The label “DMSO” and “IBRUTINIB” is present 2 times. To keep it consistent with the supplemental Figure 1B, this reviewer would suggest to remove the legend and just keep the headers, maybe adding the protein name as done in all other plots of this kind. Furthermore, the authors most likely wanted to show that the two proteoforms of BTK have different melting curves, which is fine, but it would help for clarity to state this clearly in the text.

Thank you for your attention to our figures and for making suggestions which have improved the legibility.

Action taken: we removed the excessive label and generally proportioned this figure to improve legibility. We made the size of the BTK label text larger.

Figure 3B, C, E, F, 4C: Again, here the “membership” 1 and 2 is also represented directly in the header of each plot. This might be confusing. Please just use the header which is the most obvious information readers will look at.

Action taken: we removed the membership: 0 notation for protein aggregated results for legibility and added annotation details in the headers. Further, to guide the readers, we re-proportioned and added a dividing line to show distinctions between the BRAF and YES1 parts of the figure.

Page 19, line 440: The authors rightly mention that TPP requires a substantial amount of instrument time for in-depth analysis. Would the strategy presented involving proteoforms also work with the more straightforward and faster new PISA approach (10.1021/acs.jproteome.9b00500) in which the different temperature fractions are combined before LC-MS analysis? Here, instead of 10 temperature fractions per sample, only one sample has to be analysed.

Action taken: We have been performing PISA assays and optimizing the analysis internally, although we feel the work requires additional optimization to be generalizable and replicable. We have amended the text in the discussion section (additions are noted as underlined text):

In general, our proteoform inference requires in-depth peptide detection, as well as significant instrument time and resources, which could potentially be improved by method optimization. Also, it is inherent to mass spectrometry proteomics that purely technical differences could

change the identification and quantification of peptides, which will impact downstream clustering analysis and proteoform group assignment. Technical aspects are also especially important given the fragility of native proteins, where many factors are critical for recapitulating natural drug interactions, which leads to broad technical challenges in replication across many methods of drug-target deconvolution^{1,2}. Although we opted to quantify complete melt curves, which enables a more cautious interpretation of technical aspects and regularity of melting behavior, alternative melting quantification approaches such as PISA³ could support similar analysis while tailoring for high-throughput applications.

Reviewer #1 (Remarks on code availability):

I was not able to access and download the code. The authors state that a copy is provided in the submission, but this was not found either.

Reviewer #2 (Remarks to the Author):

Leo et al. present results of TPP experiments attempting to analyse ibrutinib off-targets by exploring proteoform-level deconvolution. The paper is nice contribute to the literature on using proteoform level analysis to gain deeper insights. Whilst I'm generally supportive of the manuscript it felt somewhat rushed in places and perhaps a bit preliminary. I have several comments which I think will help improve the manuscript.

1) To address my comments around feeling rushed, please look through your figures carefully Fig. 2B has no y-axis ticks, in this figure the sizing is also different and style changes throughout. Harmonising this will make it much easier to follow. Fig. 3 the figure are out of order and generally hard to follow logically. Fig. 5 has many axis labels missing and general spacing could be improved. Fig. 6 again sizing is an issue and panel A is confusing the y-label is labelled "value" which isn't very helpful and if this is a proportion (which is how I read the legend) then why are the values over 1?

Thank you for the helpful comments, which we feel have improved the figure legibility. The axes are in fact representing proportion. Indeed, there were some samples with a value above 1, where cells were more viable in treatment conditions than DMSO conditions. We see this as expected, given that these are not immortalized cell lines and some biobanked clinical samples can have reduced baseline viability and therefore variability when cultured *in vitro*. Also, it should be considered that ibrutinib is known for its unusually modest cytotoxicity profile in an *in vitro* setting⁴.

Action taken: Axes and proportions were restored for Figure 2B. Spacing and labeling clarity has been addressed by multiple adjustments across all figures.

2) The communication of the statistics is very confusing. Why do the author keep presenting non-adjusted results alongside adjusted results, surely the non-adjusted hits are just full of false positives?

Thank you for bringing up this important point. It was not our intention that these cases are perceived as comparable to the hits meeting a more conventional pAdj threshold, and we can see how the current framing is confusing. We mentioned these statistics where they were otherwise relevant to avoid type II error, and as input in enrichment tests that have their own FDR correction, a more detailed explanation is provided to reviewer 3 as well.

Action taken: We have amended the manuscript text to ensure that “plausible” results are more clearly defined, specifying explicitly that they may be false positives (additions are noted as underlined text):

“We used BTK results to calibrate significance levels for further result interpretation, and we proceeded to consider $p_{Adj} < 0.05$ as likely thermally impacted. Although results meeting only a $p < 0.05$ threshold could be false positives, they can not be excluded. For transparency, we noted these cases may be plausibly thermally impacted and included them in Supplementary Table 1, to ensure false negatives were not implied.”

Additionally, we have added pAdj values where they were omitted, so both statistics are presented.

3) Continuing with the statistics, I feel a careful control is missing here. Because of the clustering that happens earlier, it can be easy to overstate the statistical significance. It would be useful to see an important negative control where your compound doesn't affect any proteins. Applying the same pipeline the authors should see no proteoform level changes. The way that the analysis is currently done I'm unsure if this would be the case.

To clarify, we did include this control, by using DMSO vehicle lysates, which are clustered simultaneously during the analysis. In the pulldown experiment, this is similarly controlled using vehicle probe. In the patient cohort, we have many samples which did not receive ibrutinib treatment or which received other clinically approved treatments.

For proteoform group identification analysis, the pipeline is not specifically designed to show differences in drug binding. Differences in drug binding were identified using NPARC, after the proteoform group aggregation analysis. Although we expect that temperature differences due to the drug activity could contribute to better proteoform group detection in specific cases, this is not the main factor. In fact, in many cases thermal proteoforms did not change with drug treatment, but only a few of these results were included due to the scope of the study (for example, Supplementary Figure 11A and B depict ARAF and RAF1). Below we show TMPO, where three distinct thermal proteoforms are identified in our pipeline but where none have thermal differences linked to ibrutinib treatment.

Review Figure 1: Melting of the three distinct TMPO thermal proteoforms groups, showing curves which have clear melting behavior differences where none has been linked to thermal changes in the context of ibrutinib treatment.

Our first paper introduced the methodology in whole cells without any drug treatment, and showed detection of different thermal proteoforms. We expect that these distinctions would be detected in all experimental settings that include cell-derived/non-synthetic proteomes, because protein function depends on interactions, cleavages, post-translational modifications, and other molecular processes.

4) Replication is not always clear, it useful to see explicitly in figures what the number of biological vs technical replicates was.

Thank you for this comment, although we described this in the main text it was inadvertently omitted from the figure legends.

Action taken: Figure legends have now been adjusted to specify replicates.

5) It would be useful to see these results alongside results without proteoform level deconvolution (perhaps this is there but maybe not signposted well?). Then it would be easier to see the benefits of this proteoform level deconvolution.

Thank you for this comment, we agree that showing these cases are important. We believe that unfortunately our formatting was insufficient to highlight where these cases are described in the manuscript.

Action taken: We have made this more prominent visually in Figure 3, where these cases were highlighted.

6) The prior work on this topic seems a little sparse, the authors should make clear they are applying the methodology of their previous paper. It doesn't read so generously in this current paper.

Thank you for pointing this out, this was not our intention.

Action taken: We have added additional clarifying text in the introduction (added text is indicated with underline formatting):

Many methods for global proteoform inference have been developed, including using abundance profiles⁵⁻⁷ and thermal proteome profiling (TPP)⁸. TPP is a powerful method for systematic detection and annotation of functional aspects of the proteome that uses a series of temperature treatments to resolve proteins based on their thermal stability^{9,10} which we have previously extended for proteoform identification⁸. The unique thermal stability of proteoforms can e.g. arise from different sets of PTMs, alternative splicing or proteolytic processing, and interactions with proteins, DNA, RNA, metabolites or drugs, therefore TPP approaches can capture many proteoform types⁸. Furthermore, TPP can be applied to a range of biological systems and used to analyze proteoforms in their natural contexts. This current work is an extension of this previously developed methodology.

7) The follow is a bit preliminary. I would be keen to see an assay on a protein where ibrutinib supposedly binds to one proteoform but not the other. Can you express both proteoforms and confirm this?

Indeed, this would be an interesting alternative strategy for validation. Unfortunately, because the peptide results are cumulative representations of a mixture of proteoforms,

they are not necessarily structurally distinct sites which could be used to guide a synthesis or affinity reagent approach. Our pulldown experiment attempted to address this systematically, but there are also other ways in which this could be investigated further using experimental methods.

For BRAF, we had hypothesized that the proteoform group difference reflected interactions, due to specific interaction sites which were excluded from the stabilized proteoform group. To examine this, we used full length purified BRAF and performed a kinase activity dose response assay, confirming ibrutinib can inhibit the activity of BRAF (Review figure 2). In the context of ATP, ibrutinib was not competitive - the same dose effect was seen with ATP alone or in the context of ibrutinib (Review figure 3), suggesting that despite the impact on the kinase activity, BRAF is not inhibited by interaction with its ATP binding site. Unfortunately, we could not observe different bands based on dimerization, which would require a native gel or other complex validation where we are not technical specialists. But beyond this limitation, this data supports our conclusions, and indicates a novel mechanism of BRAF inhibition which may have additional clinical utility beyond CLL. To conclusively determine the binding site and further details of the mechanism, this should be explored in follow up studies and with input from structural biology specialists, which is currently out of the scope of this study.

We are open to adding these results to supplementary if it is recommended by the reviewers or editor.

Review Figure 2: Kinase glo assay depicting dose-dependent functional inhibition of BRAF, both wild-type and the oncogenic V600E mutant.

Review Figure 3: ATP competition assay, showing a western blot and the band quantification. The experiment demonstrates dose-dependent thermal stabilization of BRAF with concurrent 20 μ M ibrutinib, with stability onset at close to the minimum physiological ATP levels sufficient for energy metabolism in cells ¹¹.

Reviewer #3 (Remarks to the Author):

In their manuscript "Proteoform-level deconvolution reveals a broader spectrum of ibrutinib off-targets", Leo et al., describe the application of their recently developed method (Kurzawa et al, Nat. Chem. Biol., 2023) to study the effects of ibrutinib on CLL cells at "proteoform-level" using thermal proteome profiling. Their method represents exciting developments in the field and I believe their new application could be an important step to promote future developments in this direction. However, there are several critical components that need to be revised.

Thank you for your insightful comments and for your attention to the manuscript. We appreciate the general support of the manuscript and the appreciation of the potential to contribute to the field. We have addressed the comments in our response below.

Major issues:

1. Terminology

The term "proteoform" has been narrowly defined to represent the exact combination of amino acid sequence and post-translational modifications (Smith & Kelleher et al., Nat Methods, 2013). Within the top-down proteomics community, which measures intact proteins, this has become an established term.

The term is also used within bottom-up proteomics, which measures mixtures of digested proteins, but typically only in introductions and discussions: Intrinsically, bottom-up proteomics can not measure proteins or proteoforms directly, but only conduct inference based on the identified peptides. This already poses a problem for the identification of canonical proteins when only shared proteins are present. In this case, the term "protein group" is used to attribute for the ambiguity.

I was confused that the authors decided to use the term "proteoform" throughout their manuscript, as it is impossible to ensure by bottom-up proteomics that the identified peptides correspond to 1 to N specific proteoforms. I found it even more confusing that in their original method paper (Kurzawa et al, Nat. Chem. Biol., 2023), they proposed a very suitable terminology themselves, calling those events "inferred functional proteoform groups". Since this subfield associating function to proteoform groups is still very young, I believe it is critical to establish very clean terminology.

Thank you for pointing this out, we certainly can see the importance of this distinction and realize how this could be a critical clarification which avoids confusion. The source of the "functional proteoform group" terminology was Isabell Bludau, who used it to describe grouped peptide variation by abundance and inter-sample correlation ⁷. This manuscript deviates from this earlier work and our own previous work, mostly that it does not use larger sample groups as a criteria to define proteoforms. But given our previous use of the terminology, for clarity, we decided to continue to use it, while adding additional clarification about the source of this term. We have now changed the wording to either "functional proteoform group", "inferred proteoform", or a variation that describes the methodology. We also highlighted this previously defined terminology in our introduction and clarified the context of our analytical methods.

Action taken: The terminology has been adjusted throughout. Specifically of note, the introduction now includes:

In the field of mass spectrometry (MS)-based proteomics, both top-down and bottom-up approaches have been used to conduct proteoform analysis without a need to pre-define variants. For example, top-down proteomics has been employed to assemble human proteoform reference maps^{12,13}; however, this method currently has limitations in depth and throughput¹³. To address this issue, bottom-up proteomics uses digested peptide libraries to achieve the broadest range of identifications. This makes it suitable for global proteome detection approaches¹⁴, but at the cost of inferring proteoforms rather than identifying them directly. The term "functional proteoform group" specifies these inferred proteoforms, where data supports a proteoform-level distinction but does not necessarily demonstrate a chemically unique, specific proteoform ⁷.

1.1: The authors should ensure that their originally proposed term "proteoform group" is used throughout the manuscript where applicable. The term "proteoform" should never be used when results could be ambiguous.

Action taken: we have corrected the terminology and checked the manuscript.

1.2: In the introduction, the authors should briefly establish the definition and relationship between "proteoform", "proteoform group" and the challenges related to classical protein inference and grouping in bottom-up proteomics.

Action taken: we have corrected the terminology and checked the manuscript.

2. Proteoform identification in ibrutinib treated cell lysates

2.1: "Although BTK_1 was the only BTK proteoform that met the $p_{Adj} < 0.05$ NPARC test significance threshold, BTK_2 was significantly shifted based on a $p < 0.05$ threshold."

This statement implies statistical significance for BTK_2, however, only if multiple-testing correction is ignored. This is not appropriate and instead, the authors should report the multiple-testing corrected value for BTK_2 as well. They can still do all comparisons and discussions downstream even if BTK_2 is not a significant hit.

2.2: "We used BTK results to calibrate significance levels for further result interpretation, and we proceeded to consider $p_{Adj} < 0.05$ as likely thermally impacted, and $p < 0.05$ as plausibly thermally impacted."

Multiple-testing corrected and uncorrected thresholds should never be mixed. Instead the authors should report the corresponding more liberal multiple-testing corrected p-value for plausibly thermally impacted hits.

Thank you for bringing up this important point. It was not our intention that these cases are perceived as comparable to the hits meeting a more conventional p_{Adj} threshold, and we can see how the current framing is confusing.

We would like to offer a more in-depth rationale for using multiple statistical measures, which we see as justified when comparing melt curves. In this study, we balanced the need to rigorously define positive results while also avoiding false negatives, and we saw these aspects as equally important. Therefore, although we agree that multiple testing correction thresholds are important when defining significance, we disagree that these statistics alone are all that is needed to meaningfully interpret results. For

example, for cases such as BTK and BRAF, using only an FDR threshold could falsely suggest proteoform group specificity, which would be far more notable but which may or may not be accurate. Using the p-value as a descriptive statistic improves contextualization of implicitly uncertain results (as defined based on FDR) using a parameter that considers the data for each comparison individually, to avoid adding risk of type II errors which are quite skewed at high detection depth when relying solely on FDR ¹⁵. This provides adequate transparency in cases where we lack the confidence to conclude a melting difference, but where we can not imply there is no difference. Therefore, we have continued to include these results in the main summary table, alongside comprehensive statistical outputs from the NPARC analysis.

When performing the interaction network and CORUM complex tests, we used both adjusted and unadjusted levels as input thresholds, which are separately processed and disclosed for transparency in the ORA table (Supplementary table 3). While the enrichment tests may have had false positives included in the initial input, it's important to note that these are conservative tests that also apply FDR corrections again at the enrichment level, and that these tests are independently applied to contextualize the more robustly defined melting hits. The ORA test we used considers overlap likelihood ¹⁶, accounting for multiple gene symbol instances and background likelihood. This approach is selective with an extremely low false positive rate, compared to conventional enrichment methods based on rank ¹⁷.

Action taken: We have amended the manuscript text to ensure that “plausible” results are more clearly defined, specifying explicitly that they may be false positives (additions are noted as underlined text):

“We used BTK results to calibrate significance levels for further result interpretation, and we proceeded to consider $p_{Adj} < 0.05$ as likely thermally impacted. Although results meeting only a $p < 0.05$ threshold could be false positives, they can not be excluded. For transparency, we noted these cases may be plausibly thermally impacted and included them in Supplementary Table 1, to ensure false negatives were not implied.”

Additionally, we have added p_{Adj} values where they were omitted.

2.3: "In addition to BTK, previous studies¹⁸ have revealed that ibrutinib binds a wide range of proteins (Supplementary Table 2), and our analysis confirmed several of these (Supplementary Figure 1C)."

Supplementary Figure 1c indicates that the overlap of the SW13 and the RCH-ACH TPP datasets is 6.8%. This seems extremely low and could be a result of multiple confounding factors, e.g. a) extreme biological diversity of cell lines, b) concentration-

cell-dependent response to ibrutinib, c) high specificity of the method, but low sensitivity, d) poor technical reproducibility, etc.

The authors should put this low overlap into context and try to explain factors contributing to it.

Thank you for this comment, we can see how this is arguably a surprising result, and agree that it may be useful to review this aspect further in the manuscript. We are not aware of a head to head comparison contrasting drug targets between cell lines, although results of pull-downs separated by cell line have been published as technical supplements, which don't contradict the hypothesis that systems-level differences by lineage could exist ^{18,19}. Many chemical proteomics methods get ahead of this by optimizing for reproducibility using e.g. proteome coverage-optimized pooled libraries of many cell lines, and it's sensible to take this approach for drug development. But despite technically-oriented experimental design, other methods still frequently lack reproducibility for undetermined reasons (linked to many factors which could affect the sometimes fragile integrity of native proteins in the assays) ^{1,2}. For example, comparing modern chemical proteomics methods such as kinobeads profiling with kinase activity assays demonstrates only weak correlation between results from each method (approximately $R = \sim 0.3$ overall) ^{1,2}. We hope that by introducing our methods, we might improve reconciliation between technical and biological differences, and that our characterisation of ibrutinib will help support future work that scales up this concept to have the technical power to explore this.

In contrast to approaches using cell line pools curated to represent pan-kinomes, our cell lines had particularly relevant biological differences: most notably, the presence and activity of on-target BTK and similar kinases. We expect that target availability has some impact on dose responses of off-targets for competitive interactions, although further work should explore this hypothesis more systematically in larger cell line libraries and ideally while accounting for dose effects. Additionally, in the context of proteoforms, amino acid sequence-level proteoforms are considered to be highly lineage specific ²⁰, and additional lineage-specific biology such as metabolite binding and protein complex composition would be broadly maintained in the lysate context in our study (for cases where they form stable interactions with proteins). In the context of ibrutinib, other work has also identified tissue-specific off-target activity, linked to metabolized ibrutinib forms which may be relevant bio-accumulation products in a clinical setting ²¹.

Action taken: We now mention this aspect in the discussion.

“Therefore, tailoring additional experimental setups to capture more realistic cellular or tissue environments may be needed to address additional targets of drug activity. These wider ranging experimental settings could also promote interpretation of context-specific results more generally, such as by reconciling the status of the targets which were not detected in both cell lines in this study or which were not confirmed from other methodology, which may enable insight into biological factors or guide improvements in technical optimization.

We also, in another part of the discussion, mention technical aspects:

In general, our proteoform inference requires in-depth peptide detection, as well as significant instrument time and resources, which could potentially be improved by method optimization. Also, it is inherent to mass spectrometry proteomics that purely technical differences could change the identification and quantification of peptides, which will impact downstream clustering analysis and proteoform group assignment. Technical aspects are also especially important given the fragility of native proteins, where many factors are critical for recapitulating natural drug interactions, which leads to broad technical challenges in replication across many methods of drug-target deconvolution^{1,2}. Although we opted to quantify complete melt curves, which enables a more cautious interpretation of technical aspects and regularity of melting behavior, alternative melting quantification approaches such as PISA³ could support similar analysis while tailoring for high-throughput applications. We hope that by introducing analytical frameworks for systematic delineation of functional proteoform groups, we might improve reconciliation between technical and biological differences, supporting future work scaling up this concept to have the technical power to explore systems-level questions.

“

2.4: Ibrutinib concentration: "RCH-ACV or SW13 cell extracts were treated with either 100µM Ibrutinib or equivalent vehicle volume (DMSO) in duplicates for 10 min at 20°C and with gentle shaking at 700 rpm."

How was the applied concentration of ibrutinib selected? Ibrutinib seems to have a very low IC50 of <0.5nM (PMC3751776). What happens if a concentration 200,000-times higher than IC50 is applied? I'm not sure if I'm missing a point here, but the authors should provide a clear rationale for how the drug concentration was selected, as this can have a tremendous effect on the context of the results and invalidate the whole concept of a study. When screening drug compounds in vivo, ideally, selected concentrations should be within a physiological scale similar to effective drug concentrations in tissue. I understand that for this study, screening was not conducted in vitro but in vivo. But such a dramatic difference in drug concentration will have a tremendous effect on off-target detection, which might be entirely absent or irrelevant in clinical settings.

The authors should demonstrate and discuss whether their results can translate to in vivo or clinical scenarios, and if not, what limitations (e.g. context-specificity of observed off-target effects) should be expected.

Thank you for your comments, we appreciate the opportunity to clarify our rationale and address your concerns. For clinical utility, ibrutinib is given to patients over long periods of chronic disease management, and in these contexts drug availability is not as simple as on-target IC50. In PK-PD studies, plasma drug concentrations (Cmax) are much higher, with observations depending on the disease indication and patient group but ranging between ~10uM ²² to ~80uM ²³. And other baseline conditions or comorbidities are known to reduce ibrutinib metabolism leading to more bioaccumulation, notably including treatment with some antibiotics ²¹.

On a technical level, this assay was performed in cell lysates, which have the benefit of lacking downstream events that change protein stability but inherently do not match physiological protein concentration and distribution in cells. Further, stability change quantification requires target saturation, not just physiologically active binding ²⁴. You may note that the shift for BTK_1 is of a small magnitude, although it is significant - this illustrates the primary benefit of target saturation, which is that minimizing available unbound protein stoichiometry achieves better confidence intervals and replication of melting and enables better technical measurements of small melt changes. Additionally, when interpreting proteoform groups, we did not want to introduce uncertainty related to incomplete saturation, where a minimal dose would be expected to introduce many

false negatives due to noise.

Review Figure 4: Isothermal dose response assay, showing ibrutinib concentration dose-response stabilization of BRAF in the presence of ibrutinib performed at 50C. We are open to adding these results to supplementary if it is recommended by the reviewers or editor.

To accurately assess IC₅₀ of off-targets, traditionally other tests are performed such as isothermal dose response profiling, which we demonstrate for BRAF (IC₅₀ = ~6μM, **Review Figure 4**), and if they exist, these thermal results can also be compared with an activity-based kinase assay (**Review Figure 2**). Isothermal dose response assays are targeted follow up experiments, not proteome-wide, because they depend on expected melting temperature information for each detected protein. The 2DTPP method has been developed to quantify both melt change and dose response simultaneously at proteome level²⁵, however, this method is not yet robustly amenable for proteoform detection applications due to the need to prioritize either drug or temperature differences experimentally when designing the TMT set layouts. Essentially, a traditional 2DTPP based assay would not be able to detect melt curves, instead only concentration curves per temperature, which would achieve a very different type of “functional proteoform” detection using drug binding affinity differences while remaining blind to baseline melting behavior. We anticipate this approach would be less reliable and more

resource intensive, although of course it could be developed further for future applications to address these questions of dosing dependent stabilization.

3. Using proteoform data to detect effects of drug binding on protein-protein interactions and complex formation

3.1: "Protein-protein interactions are highly dependent on cell lineage⁵³, and we observed that the baseline thermal stabilities of complex-associated proteoforms and magnitude of drug-induced thermal changes were not uniform between cell lines."

I think this is an extremely critical and potentially impactful part of the study. However, I'm worried about the comparison and interpretation of the results. If I understood correctly, differential analysis was conducted of the two cell lines separately of ibrutinib vs DMSO. Then, the two lists of significant hits were compared to assess potential differences. Unfortunately this will exclude hits with sparse coverage across all samples and call those differential, which are at the border of significance, right?

One approach for a more statistically meaningful comparison would be to employ an "interaction model" (e.g. as implemented in LIMMA) to compare (SW13_ibrutinib / SW13_DMSO) / (RCH-ACV_ibrutinib / RCH-ACV_DMSO). The authors should reformulate their differential comparison to ensure that differentially identified interactions are really significant and not observed due to technical artefacts.

Thank you for bringing this up, we can see the importance of making this analysis much more clear than our current presentation. To clarify, all statistical melting comparisons using NPARC were filtered within the function to consider minimally 40 points for comparisons using both cell lines, representing a full duplicated melt curve set of quantification points for each separate treatment group. Within each lineage-specific ORA analysis, the detection background was the set of IDs which obtained a valid statistical result in these previous NPARC tests, regularizing the ORA to properly consider chance by how many times a gene symbol is present and also distinguishing between detected and not significant (impacts the ORA) versus not detected (not considered in the ORA). The analysis evaluating the complexes was performed separately in order to account for this background aspect, and our initial hypothesis was that valid biological differences would be expected to also include some results where a complex component was missing in the data. If we included both together, the statistical setup would not be able to accurately consider background and might also exclude

confirmatory results.

For the complexes depicted in supplementary figures 2-6 (the lineage-specific protein complex results), we included plots of all melting curves, demonstrating that there are not many missing IDs despite being permitted in theory. Among these, LCK is the exception in that it was not detected in any form for SW13. This was shown in supplementary figure 6 by an empty labeled plot to demonstrate lack of quantification. The complex is somewhat confirmatory as it is known as lymphocyte-specific²⁶ and linked to on-target BTK functions, and we are confident it is not an artifact of poor detection depth.

We appreciate the suggestion to use an interaction model, which we agree is an approach that could have utility in interpreting melting differences linked to lineage for specific proteoform groups. We applied this using NPARC, to remain consistent in fitting curves (given the non-linear data structure), and we would be open to add this to the manuscript if recommended. However, these interaction model results were not included in the initial manuscript because the effect of different cell line backgrounds broadly affects baseline melting at a greater magnitude and frequency than drug effects, leading to extremely high pAdj estimates. Despite these statistical limitations, the results generally align with the other analyses. The top 10 differential melting results between treatment conditions are presented below:

id	rssDiff	fStat	pVal	pAdj
MAP2K7_1	80,1183057	25,8526791	1,287E-05	0,04573241
MAP2K5_1	29,1440788	25,3799848	1,4279E-05	0,04573241
MAP2K5_2	35,2095567	25,2155798	1,481E-05	0,04573241
CSK_2	27,6973812	21,6831475	3,4094E-05	0,07896281
ACADM_1	14,1490617	18,279263	8,4699E-05	0,15693102
BRAF_1	24,8740428	16,3963228	0,00014817	0,20049737
DCUN1D5_1	11,551587	16,3246936	0,0001515	0,20049737
ALDH7A1_2	35,0809705	15,2036662	0,00021642	0,25061488
MSRB2_2	25,8418904	14,7355171	0,00025257	0,25998321
ALDH7A1_1	28,7774367	13,9159506	0,00033383	0,30925882

Action taken: We have added additional detail to the methods section when describing input for NPARC and ORA analysis.

“Differential melting curve analysis was performed using NPARC, as previously published and described in the context of this analysis^{8,27}. Each comparison was filtered to consider only full and duplicated melting curves in each treatment group.”

“Protein complex enrichment was performed using ORA with the Comprehensive Resource of Mammalian Protein Complexes (CORUM) database, also retrieved in the R package as the Homo Sapiens “network_CORUM” enrichment database. The full list of unique gene symbols that had 8 full melting curves were used as reference genes, and gene symbols with significant thermal melting changes as identified with NPARC were used as the hits input. Significance was determined using the false discovery rate (FDR) method Benjamini-Hochberg, “BH”, with a threshold of 0.05. A minimum number of 3 gene symbols in each CORUM complex category was required when assembling the input database. Multiples of gene symbols representing different functional proteoform identities were considered separately, in both the enrichment list and the background list, enabling the analysis to consider the background chance of a gene symbol’s random identification accurately, and to prioritize cases where proteoform groups under the same gene symbol category share meaningful engagement across multiple measured protein contexts.

In the main text, we have also added:

“To examine this further, we repeated the ORA tests within each cell line separately (Supplementary Table 3) to allow the ORA to have proper statistical input for background detection and because cell line backgrounds affected baseline melting more than drug effects. “

”

4. Validation by peptide resolved pulldown experiment

4.1: "Results were considered significant if they were replicated in two out of three ibrutinib treated preparations without DMSO detection, and results detected in both ibrutinib and DMSO pulldown samples were considered if they were replicated in at least two preparations and also had intensities that met a paired t-test threshold of $p < 0.05$ "

The authors should provide multiple-testing corrected p-values and thresholds for pull downs as well. It is acceptable if this results in a large cutoff, e.g. 20-50% FDR, but it is critical that readers have the opportunity to judge the confidence of the results.

Thank you for your suggestion, we have now added the multiple-testing corrected p-values for the pull-down experiments as well. These results are all approximately equal to one when they are adjusted for multiple testing, however exclusion of these results would not change the count of 7 shared hits. Inadvertently, although LYN_1 is shown in the plot, this result and other similar cases were in fact excluded from Figure 4A due to an inadvertently high significance threshold, which has been adjusted now to reflect pulldown hits as results in at least two ibrutinib preparations without DMSO detection, excluding p value linked hits.

Action taken: For simplicity, we removed the description of p value linked hits. Adjusted p values appear alongside the results in the table (Supplementary table 5).

5. Proteoforms in CLL patients vary with treatment

5.1: "Using this cohort of 68 patients profiled using HiRIEF LC/MS proteomics, peptides were summed to proteoforms and considered by their relative abundance, an independent metric not linked to their thermal behavior²⁷."

This part would benefit from more explanations. To make sure that I understand this correctly: Across the 68 CLL patient cohort, an F-test was conducted testing the intensities of those peptides grouped to a functional proteoform group by TPP, against the intensities of a randomized selection of peptides mapping to the same gene identifier measured in the 68 CLL patient cohort.

Would we not expect the randomized selection to have intrinsic bias, as they could not uniquely associated to a functional proteoform group by TPP? Presumably, those could be shared peptides that intrinsically have higher abundances?

Also, I struggle with the quantitative comparison of different peptides: Quantitative protein inference algorithms such as MaxLFQ work because they can operate on shared sets of peptides. Here, the comparisons are only conducted on sets of unique peptides. An appropriate comparison would be to first normalize raw peptide intensities against matching healthy tissue, which is provided by most label-based studies. Maybe the authors have done this, but from their methods section, this was not clear, as they write "The PSM values were VSN normalized by set. Then, the PSMs were summed by the following 3 parameters", which suggests label-free spectral counting. In absence of matching healthy tissue samples, the authors could redo the analysis with any non-CLL

samples that have similar proteome coverage to demonstrate that indeed proteoform groups drive differential abundance and not the selection of peptides.

The authors should clarify the methodological assumptions and demonstrate that peptide intensity and not peptide selection is driving differential proteoform group abundance.

Thank you for raising this point, we apologize for the wording here which we can see was not adequate to describe the analysis. Describing “random”, we were referring to random re-assignment of membership identities for the same peptides - we did not select random peptides, rather the analysis was performed with the same peptides that could be assigned as a functional proteoform group component. The same peptides per group were assigned to memberships according to the same properties of the parent distribution, using the same number of peptides per membership from the same peptide quantifications. This approach was meant to mimic an unreasoned proteoform assignment logic, to catch cases where intra-gene symbol variability was high and susceptible to misinterpretation when subsetted. Here, this analysis was only used as a null distribution in f-tests to compare variance, rather than abundance. For differential abundance tests, the 4 patients who were ibrutinib treated were compared to patients who received other treatments or were untreated at time of sampling in the 68 patient cohort.

Action taken: We have now clarified this in the manuscript:

“Then, the PSMs were summed by the following 3 parameters: by gene symbol, by functional proteoform group (for matched peptides), and by “random” functional proteoform group (shuffling only the membership identification, also for matched peptides). When selecting “random” peptides, the functional proteoform group membership identifier was scrambled for the same set peptides, maintaining the proportion, variability, quantifications, and parent distribution properties for matched peptides per gene symbol, to be as comparable as possible to the proteoform summed data. After summation, the data was log transformed and median centered. The “random” peptide data set was used as a null model in f-tests to evaluate proteoform variation and underlying noise in peptide-level data.”

Minor issues:

- All figures: Please indicate statistical significance on all melting curves.

- Fig 1a: I don't understand the rationale behind Fig. 1a. Please either extend the description to guide the reader or better, replace this figure with a schematic of the experimental design.
- Fig 1d: This figure is way too small.
- Fig 1e: This is an extremely non-standard figure. Is this just a barplot to display significance? This could be omitted or illustrated in a much easier way.

Thanks for your attention to the figures and for your advice to improve their legibility. We have made some cosmetic changes in line with these comments. Regarding figure 1e, we understand that it might be sub-optimal, it is a “lollipop plot” more commonly used in other contexts.

Action taken: We removed this figure during our reorganization, now only mentioning the values in the text.

Reviewer #3 (Remarks on code availability):

The R-code is provided as an archive including a brief README file.

1. Reinecke, M. *et al.* Chemical proteomics reveals the target landscape of 1,000 kinase inhibitors. *Nat. Chem. Biol.* **20**, 577–585 (2024).
2. Rudolf, A. F., Skovgaard, T., Knapp, S., Jensen, L. J. & Berthelsen, J. A comparison of protein kinases inhibitor screening methods using both enzymatic activity and binding affinity determination. *PLoS One* **9**, e98800 (2014).
3. Gaetani, M. *et al.* Proteome Integral Solubility Alteration: A High-Throughput Proteomics Assay for Target Deconvolution. *J. Proteome Res.* **18**, 4027–4037 (2019).
4. Herman, S. E. M. *et al.* Bruton tyrosine kinase represents a promising therapeutic target for treatment of chronic lymphocytic leukemia and is effectively targeted by PCI-32765. *Blood* **117**, 6287–6296 (2011).
5. Bamberger, C. *et al.* Deducing the presence of proteins and proteoforms in quantitative proteomics. *Nature Communications* vol. 9 Preprint at <https://doi.org/10.1038/s41467-018->

- 04411-5 (2018).
6. Dermit, M., Peters-Clarke, T. M., Shishkova, E. & Meyer, J. G. Peptide Correlation Analysis (PeCorA) Reveals Differential Proteoform Regulation. *J. Proteome Res.* **20**, 1972–1980 (2021).
 7. Bludau, I. *et al.* Systematic detection of functional proteoform groups from bottom-up proteomic datasets. *Nat. Commun.* **12**, 3810 (2021).
 8. Kurzawa, N. *et al.* Deep thermal profiling for detection of functional proteoform groups. *Nat. Chem. Biol.* (2023) doi:10.1038/s41589-023-01284-8.
 9. Martinez Molina, D. *et al.* Monitoring drug target engagement in cells and tissues using the cellular thermal shift assay. *Science* **341**, 84–87 (2013).
 10. Savitski, M. M. *et al.* Tracking cancer drugs in living cells by thermal profiling of the proteome. *Science* vol. 346 Preprint at <https://doi.org/10.1126/science.1255784> (2014).
 11. Greiner, J. V. & Glonek, T. Intracellular ATP Concentration and Implication for Cellular Evolution. *Biology* **10**, (2021).
 12. Tran, J. C. *et al.* Mapping intact protein isoforms in discovery mode using top-down proteomics. *Nature* **480**, 254–258 (2011).
 13. Smith, L. M. & Kelleher, N. L. Proteoforms as the next proteomics currency. *Science* vol. 359 1106–1107 Preprint at <https://doi.org/10.1126/science.aat1884> (2018).
 14. Aebersold, R. & Mann, M. Mass-spectrometric exploration of proteome structure and function. *Nature* **537**, 347–355 (2016).
 15. Rothman, K. J. No adjustments are needed for multiple comparisons. *Epidemiology* **1**, 43–46 (1990).
 16. Liao, Y., Wang, J., Jaehnig, E. J., Shi, Z. & Zhang, B. WebGestalt 2019: gene set analysis toolkit with revamped UIs and APIs. *Nucleic Acids Res.* **47**, W199–W205 (2019).
 17. Buzzao, D., Castresana-Aguirre, M., Guala, D. & Sonnhammer, E. L. L. Benchmarking enrichment analysis methods with the disease pathway network. *Brief. Bioinform.* **25**,

- (2024).
18. Bantscheff, M. *et al.* Quantitative chemical proteomics reveals mechanisms of action of clinical ABL kinase inhibitors. *Nat. Biotechnol.* **25**, 1035–1044 (2007).
 19. Lanning, B. R. *et al.* A road map to evaluate the proteome-wide selectivity of covalent kinase inhibitors. *Nat. Chem. Biol.* **10**, 760–767 (2014).
 20. Melani, R. D. *et al.* The Blood Proteoform Atlas: A reference map of proteoforms in human hematopoietic cells. *Science* **375**, 411–418 (2022).
 21. Rood, J. J. M. *et al.* Extrahepatic metabolism of ibrutinib. *Invest. New Drugs* **39**, 1–14 (2021).
 22. de Jong, J. *et al.* Ibrutinib does not have clinically relevant interactions with oral contraceptives or substrates of CYP3A and CYP2B6. *Pharmacol Res Perspect* **8**, e00649 (2020).
 23. Scheers, E. *et al.* Absorption, metabolism, and excretion of oral ¹⁴C radiolabeled ibrutinib: an open-label, phase I, single-dose study in healthy men. *Drug Metab. Dispos.* **43**, 289–297 (2015).
 24. Jafari, R. *et al.* The cellular thermal shift assay for evaluating drug target interactions in cells. *Nat. Protoc.* **9**, 2100–2122 (2014).
 25. Childs, D. & Kurzawa, N. Introduction to the TPP package for analyzing Thermal Proteome Profiling data: 2D-TPP experiments. *andersvercelli.com* (2016).
 26. Zhou, J., Zhang, Q., Henriquez, J. E., Crawford, R. B. & Kaminski, N. E. Lymphocyte-Specific Protein Tyrosine Kinase (LCK) is Involved in the Aryl Hydrocarbon Receptor-Mediated Impairment of Immunoglobulin Secretion in Human Primary B Cells. *Toxicol. Sci.* **165**, 322–334 (2018).
 27. Childs, D. *et al.* Nonparametric Analysis of Thermal Proteome Profiles Reveals Novel Drug-binding Proteins. *Mol. Cell. Proteomics* **18**, 2506–2515 (2019).

Nature Communications manuscript NCOMMS-24-01474B Rebuttal

REVIEWERS' COMMENTS

Reviewer #1 (Remarks to the Author):

The authors have addressed all my comments satisfactorily.

I have also been asked to comment on the concerns of reviewer #3.

Reviewer #3 raised valid technical concerns regarding the use of the term proteoform, statistical data analysis and presentation of the data as well as the use of a very high drug concentration in the experiments. In my view, the authors have very carefully and convincingly addressed all comments raised by Reviewer #3, also by adding new results. Thus, in my opinion, the manuscript is now considerably improved and in an appropriate state for publication.

Thank you for your positive feedback and support of our manuscript.

Reviewer #2 (Remarks to the Author):

The manuscript is greatly improved in response to the revision and appears to be acceptable. I don't need to see a revised version but I note there is heavy reliance on technical duplicates, which on one hand results in reduced power and on the other hand could underestimate biological variability. I appreciate from a time and cost perspective this may be the only option but the limitation should at least be mentioned in the discussion, so those doing similar experiments can understand why their might be inconsistencies.

Thank you for this comment, we agree that the discussion could have more detail on this topic. We added the following statement to our discussion, the added text is highlighted in green color:

“Despite improving quantification of melt differences, focusing on full melt curve quantification limited our study's scope. Results from two cell lines indicated surprising biological variability, reducing statistical power, and the true biological variability between additional contexts may be higher. Future work could provide a more comprehensive understanding of variability by addressing inconsistencies when replicating experiments or comparing results across different studies to improve the robustness of our findings.”